# Phenology is the dominant control of methane emissions in a tropical non-forested wetland

Carole Helfter [1✉], Mangaliso Gondwe [2], Michael Murray-Hudson [2], Anastacia Makati [2], Mark F. Lunt[3], Paul I. Palmer [3,4] & Ute Skiba[1]

Tropical wetlands are a significant source of atmospheric methane ($CH_4$), but their importance to the global $CH_4$ budget is uncertain due to a paucity of direct observations. Net wetland emissions result from complex interactions and co-variation between microbial production and oxidation in the soil, and transport to the atmosphere. Here we show that phenology is the overarching control of net $CH_4$ emissions to the atmosphere from a permanent, vegetated tropical swamp in the Okavango Delta, Botswana, and we find that vegetative processes modulate net $CH_4$ emissions at sub-daily to inter-annual timescales. Without considering the role played by papyrus on regulating the efflux of $CH_4$ to the atmosphere, the annual budget for the entire Okavango Delta, would be under- or overestimated by a factor of two. Our measurements demonstrate the importance of including vegetative processes such as phenological cycles into wetlands emission budgets of $CH_4$.

[1] UK Centre for Ecology and Hydrology, Penicuik EH26 0QB, UK. [2] Okavango Research Institute, University of Botswana, Maun, Botswana. [3] School of GeoSciences, University of Edinburgh, Edinburgh, UK. [4] National Centre for Earth Observation, University of Edinburgh, Edinburgh, UK. ✉email: caro2@ceh.ac.uk

Atmospheric concentrations of methane ($CH_4$), the second most important greenhouse gas after carbon dioxide ($CO_2$), have increased steadily since 2007[1–3] after nearly a decade of zero growth[4] but the causes for this renewed increase are not fully understood. Possible explanations include (1) an unidentified increase in $CH_4$ emissions from anthropogenic sources such as oil and natural gas[5], (2) a reduction in $CH_4$ destruction due to changes in the oxidative capacity of the atmosphere[6,7], (3) an increase in biogenic emissions inferred from a shift in the isotopic signatures of atmospheric $CH_4$[8] and (4) a decrease in biomass[9] burning reconciling the combined effects of (1) and (3). The isotopic shift supports the idea of an increase in biogenic emissions, particularly from the tropics[10] that contribute ~65% of the global $CH_4$ budget. Estimates of global emissions have large uncertainties, with bottom-up (BU) budgets (inferred from process-based models and inventories) over-estimating top-down (TD) budgets (estimated through inversion modelling constrained by atmospheric $CH_4$ concentration measurements) by ca. 30%[11]. Wetland budgets are particularly uncertain: for example, during the 2000–2009 period, global estimates were 9% larger for BU than TD budgets[12], but this figure was recently revised and BU are currently 22% smaller than TD budgets[11]. Despite these uncertainties, the current estimate is that emissions from tropical wetlands have increased by ~10% between the 2000s and 2017, and account for ca. 20% of the global $CH_4$ budget. These BU and TD estimates provide little information about the underlying processes so there is an urgent need to consider understudied emission mechanisms and processes that might help reconcile estimates from the two approaches.

This is particularly timely, given that revisions of conventional model parameters such as wetland extent[13–15] and temperature-sensitivity of methanogenesis[16] fail to reconcile BU and TD estimates or explain recent inter-annual variations in emissions. Net fluxes of $CH_4$ to the atmosphere result from complex and sometimes competing processes, which underpin production, oxidation and transport, and the magnitude and temporal dynamics of these terms are intimately linked to vegetative processes and growth cycles. For example, the availability of C for $CH_4$ production in soils, either from plant litter or photosynthates, can control $CH_4$ production and contribute to the modulation of daily to seasonal emissions in ecosystems spanning subarctic to subtropical latitudes[17,18], and the level of methane oxidation in soils has been shown to be larger in vegetated soils due to plant-mediated oxygenation of the rhizosphere[19,20]. Plant-mediated transport of $CH_4$ can be the dominant transport mechanism[21], but the efficiency of this pathway can be species-dependent and variable[22–24].

Recent studies in the tropics have shown that trees can be substantial sources or pathways of $CH_4$ to the atmosphere[25,26], but less is known about the role of emergent macrophytes in non-forested tropical wetlands, which account for 20–37% of the global land surface of vegetated wetlands[27–29]. Whilst parameters such as soil C and inundation are commonly used in process-based models of $CH_4$ emissions, transport, and particularly plant-mediated transport of $CH_4$ is a relatively poorly represented pathway. Regional and global estimates of wetland $CH_4$ emissions from models with and without explicit treatment of the transport pathway can vary by up to a factor two[30], and uncertainties on emissions from the data-poor tropics are particularly large.

In this work, we report on three years of measurements of land-atmosphere exchange of $CH_4$ in permanent and seasonally flooded wetlands in the Okavango Delta, Botswana, and demonstrate that net $CH_4$ emissions are broadly controlled by seasonal changes in regional hydrology and, particularly, by the associated phenological cycle of the wetland vegetation. Furthermore, the marked diel cycles in methane emissions observed at the perennial swamp study site, challenge the common practice of upscaling daytime fluxes to higher temporal statistics. We also demonstrate that the seasonality in $CH_4$ fluxes measured on the ground is observed by satellite data and that we can reconcile annual BU and TD $CH_4$ budgets for the entire Okavango Delta. We established two eddy-covariance (EC) sites in the Okavango Delta in August 2017 to quantify $CH_4$ fluxes ($F_{CH4}$) from a seasonal floodplain and from a permanent wetland. These EC systems were still active at the time of writing and our EC dataset is, to the best of our knowledge, the longest continuous record of wetland $CH_4$ fluxes in Africa.

## Results and discussion

**$CH_4$ methane emissions at the perennial swamp**. The largest $CH_4$ emissions (0.10–0.64 $g\,m^{-2}\,d^{-1}$) in the Okavango Delta are from the perennial swamp areas, which are dominated by emergent *Cyperus papyrus* and *Phragmites australis*. Monthly median fluxes of $CH_4$ ($F_{CH4}$) measured at the Guma Lagoon papyrus swamp (18°57′ 53.01″S; 22°22′16.20″E) are linearly and positively correlated ($R^2 = 0.89$, p-value from t distribution <0.05) with monthly maximum gross primary production ($GPP_{MAX}$, "Methods") as shown in Fig. 1a. $GPP_{MAX}$, which represents the maximum photosynthetic potential of the papyrus stand, is a measure of the vigour/health of the stand at a particular time and the strongest predictor for $F_{CH4}$. $F_{CH4}$ and $GPP_{MAX}$ tend to increase with air and water temperature but these relationships are not statistically significant. This is consistent with other studies, which documented significant correlations between $CH_4$ fluxes and GPP at monthly to seasonal timescales. At such timescales, the net $CH_4$ fluxes to the atmosphere are most likely modulated by the $CH_4$ production term, which is stimulated by carbon (C) deposition in the rhizosphere from photosynthates[31–33].

We also found a positive correlation between $F_{CH4}$ and monthly estimates of enhanced vegetation index (EVI; Fig. 1b, $R^2 = 0.69$, p-value < 0.05), and a comparable correlation ($R^2 = 0.66$, p-value < 0.05) between $GPP_{MAX}$ and EVI. EVI, a proxy for leaf phenology or biomass[34] was obtained from Sentinel-2 imagery for the portion of the papyrus stand within the flux footprint of the EC tower. The phenological cycle of papyrus is complex: instead of a seasonal full stand die-back, senescence and recruitment of new shoots co-occur throughout the year, but the ratio of mature to senescing plants is variable[35]. The phenological cycle of papyrus inferred from EVI is characterised by high EVI values (green vegetation/more above-ground biomass) during summer months (December–February) followed by a gradual decline (senescence) until winter (June–August). We propose that $GPP_{MAX}$, a metric comparable to the photosynthetic capacity of Wu et al.[34] is an indicator of the maturity of the papyrus stand or of leaf ontogeny. $GPP_{MAX}$ might hence be more representative of, or biased towards, mature plants than EVI, because these are more productive/photosynthetically active than juvenile and senescent ones; as demonstrated by Wu et al.[34], leaf ontogeny is a strong predictor for GPP, but EVI cannot resolve leaf age which results in a weaker correlation.

**$CH_4$ emission mechanisms and processes in the perennial swamp**. The correlation between $F_{CH4}$ and EVI is likely an artefact of the circular dependency of $F_{CH4}$ on $GPP_{MAX}$ and $GPP_{MAX}$ on EVI, but based on the available data it is not possible to rule out an element of seasonality in the plant-mediated transport term of $CH_4$. Indeed, a small study on papyrus reported negligible $CH_4$ fluxes through the culms of juvenile and senescing individuals. By extrapolation, this could mean that the largest $CH_4$ fluxes for an entire papyrus stand occur when the proportion of mature plants reaches a maximum, i.e. when $GPP_{MAX}$, but not necessarily EVI, is at its maximum. This is consistent with the

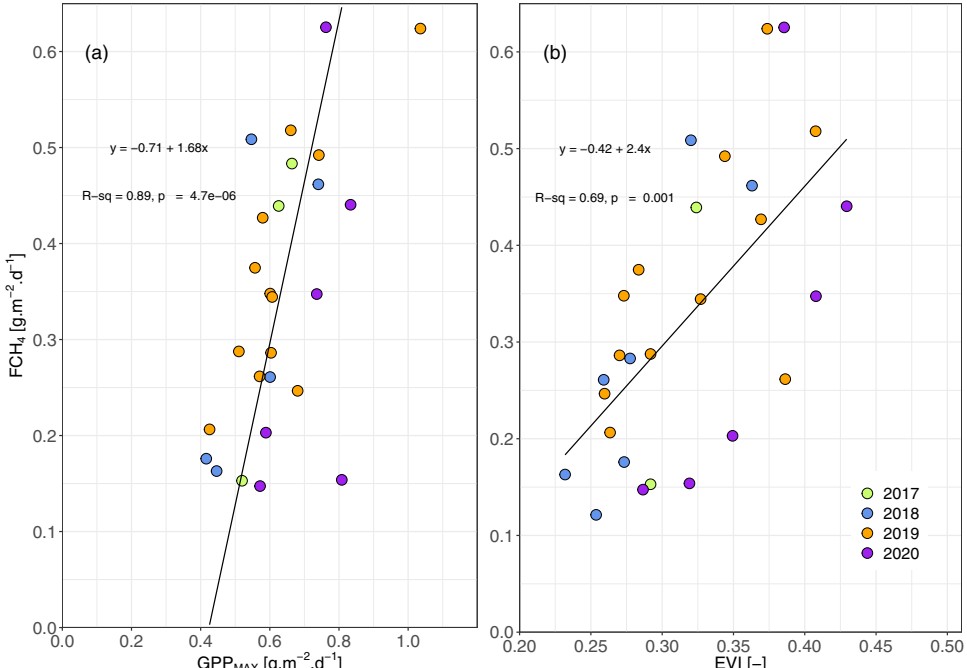

**Fig. 1 Mean monthly flux of CH$_4$ measured by eddy-covariance over *Cyperus papyrus*.** Measurements taken at Guma Lagoon (18°57′53.01″S; 22°22′16.20″E) in the permanently flooded part of the Okavango Delta from August 2017 to August 2020. Fluxes (FCH$_4$) are presented as function of (**a**) maximum gross primary productivity (GPP$_{MAX}$), and (**b**) enhanced vegetation index (EVI). The solid lines represent linear regressions (equations and t-test two-sided *p*-value given in the panels): standard error of (**a**) ±0.19 and 0.12, and (**b**) ±0.51 and 0.17 for slope and intercept, respectively.

findings of Whiting and Chanton[18], who explained the strong positive linear correlations between F$_{CH4}$ and biomass in a variety of plants, as a combination of rates of the plant-mediated organic substrate supply and plant-mediated transport. Working on the assumption that CH$_4$ is mainly lost through non-stomatal pathways such as pores in plant stems, they suggested that more biomass could equate with larger conduits for CH$_4$ transport and hence greater fluxes. In conclusion, whilst variations in physiology or plant-mediated transport efficiency are plausible, the dominant control of the seasonal cycle in CH$_4$ emissions is probably C-deposition to the rhizosphere.

We observed marked diel cycles from October to March (Fig. 2), characterised by a decrease in F$_{CH4}$ during the central daylight hours when vapour pressure deficit (VPD) was high and CO$_2$ uptake peaked. Overall, emissions of CH$_4$ were 52 ± 26% larger at night than during the day (median ± inter-quartile range of monthly night to day ratios), and this night-to-day ratio increased with increasing EVI (Extended Fig. 2). Both day and night-time fluxes were linearly correlated with EVI but the night-time slope was almost double the daytime one. Night-time fluxes also had a weak statistically significant correlation with mean air temperature, but that was not the case for daytime fluxes. The implication of these differences between night and day fluxes is that estimating daily or higher temporal budgets from daytime measurements would lead to significant underestimations. To the best of our knowledge, there is no published literature on the dominant plant-mediated transport pathway for CH$_4$ in *Cyperus papyrus*, but a limited study documented constant CH$_4$ efflux from mature culms throughout daylight hours, and negligible emissions from juvenile and senescent plants[35]. Such constant emission patterns do not fit the marked diel trends observed for most of the year, but stomatal flux regulation is a possibility. Jones and Muthuri showed[36] that the stomatal conductance in papyrus canopies exhibits a sharp early morning rise followed by partial closure around midday as VPD increases. Partial stomatal closure reduces transpiration, even when root zone water availability is high, but has a lesser impact on

photosynthesis in C4 plants[37] such as papyrus[38]. Consequently, partial stomatal closure could reduce CH$_4$ emissions as well as transpiration, while affecting CO$_2$ uptake, and by extension GPP$_{max}$, to a smaller degree. CH$_4$ fluxes through the umbels of the plants, which are comprised of hyperstomatal bracteoles and rays, and their diel cycles are however unknown. Although plausible, stomatal control of CH$_4$ fluxes in papyrus at short timescales remains speculative.

Pressurisation of the internal lacunae found in many aquatic macrophytes in response to increasing VPD and, to a lesser degree, air temperature, can give rise to convective gas flow[39], albeit with temporal cycles opposite to the ones we measured over papyrus[39,40]. Some authors have attempted to reconcile the concept of convective flow with observed trends of decreasing CH$_4$ emissions during daylight hours as venting of CH$_4$ accumulated overnight inside the plants until fluxes become limited by pressurised ventilation, or by root-shoot gas transport rather than stomatal conductance[23,31,41]. To the best of our knowledge, pressurisation has not been studied in *Cyperus papyrus*, but other members of the Cyperaceae family (e.g. *C. involucratus* and *C. eragrostis*) are known not to produce significant convective flow[39].

Diel cycles of O$_2$ fluxes through plants, and concentrations within the rhizosphere, which can decrease by 30% at night in papyrus as a result of metabolic O$_2$ demand and the absence of photosynthetic production[42], offer an additional mechanism for the measured trends. A concomitant reduction in CH$_4$ oxidation at night could shift the balance of the CH$_4$ production and consumption terms towards a net increase in CH$_4$, giving rise to diel emission trends consistent with our measurements[43]. This could also explain the strong correlation between CH$_4$ fluxes and EVI, particularly at night (Supplementary Fig. 1), as more live biomass (high EVI) could equate with higher metabolic O$_2$ demand and higher net CH$_4$ fluxes. In all likelihood, the trends in net CH$_4$ emissions result from a superposition of several processes, but our dataset cannot resolve them.

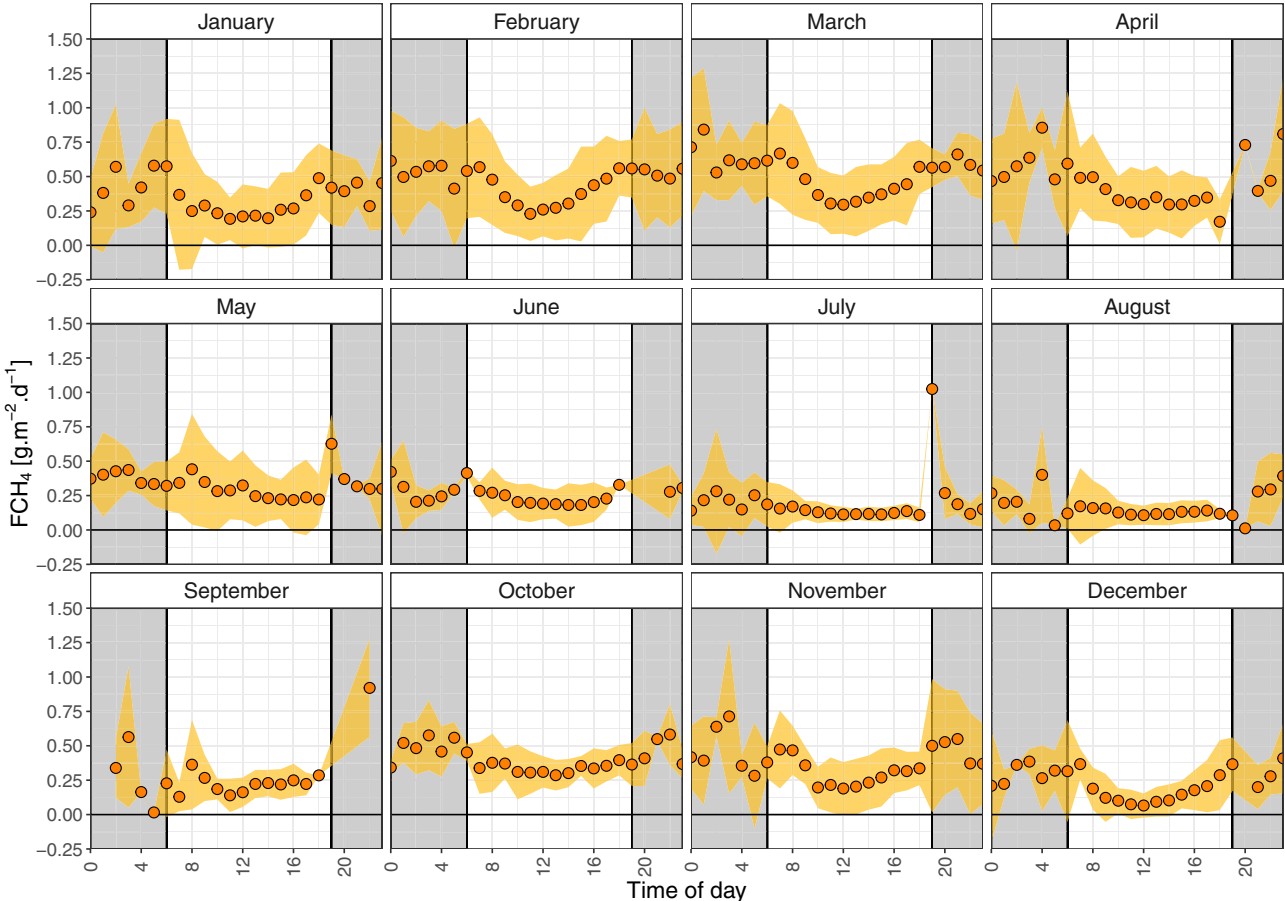

**Fig. 2 Diel cycles of methane fluxes measured by eddy-covariance over *Cyperus papyrus*.** Measurements taken at Guma Lagoon (18°57′53.01″S; 22°22′16.20″E) in the permanently flooded part of the Okavango Delta from August 2017 to August 2020. Half-hourly methane flux data points (FCH₄) were averaged to hourly values on a monthly basis using all available data for the period August 2017–August 2020 were used. The coloured ribbon represents the standard deviation of the mean and the grey rectangles symbolise night-time.

The seasonal trend seen in EC data was also observed in satellite-derived fluxes, albeit with an apparent time lag of ca. one month between EC and satellite (Fig. 3). This temporal difference is likely due to an asynchronicity between the environmental controls of the $CH_4$ fluxes observed locally (EC) and at the scale of the entire Delta (satellite-derived fluxes). Furthermore, inter-annual variations in surface properties may have impacted the seasonality of the $CH_4$ fluxes: for example, in 2019, the onset of greening and browning of the papyrus stand at Guma Lagoon (inferred from EVI, Supplementary Fig. 2) lagged other years by one to two months, which indicates a variability in the timing of vegetation development. This was not tested at the broader basin scale, but basin-wide means of both surface temperature and EVI account for 30% of the variance in estimated $CH_4$ emissions.

**Controls of $CH_4$ emission dynamics at the seasonal floodplain.** At Nxaraga seasonal floodplain (19°32′53″S; 23°10′45″E, site description in Methods), which is dominated by grasses such as *Panicum repens*, *Cynodon dactylon*, and *Sporobolus spicatus*, monthly median $CH_4$ emissions ranged from 0.006 to 0.262 g m⁻² d⁻¹ and exhibited a complex dependence on both water level (WL) and air temperature ($T_{air}$, Fig. 4). Figure 4 suggests that water level and air temperature are predictors of $F_{CH4}$, until water levels drop below a critical threshold, at which point the these variables uncouple from $F_{CH4}$. In 2017 and 2018, $CH_4$ emissions increased from winter to summer as $T_{air}$ increased ($R^2 = 0.06$, $p$-value $= 0.4$) and WL

decreased ($R^2 = 0.43$, $p$-value $= 0.05$). Whilst the correlation between $F_{CH4}$ and $T_{air}$ was not statistically significant (Fig. 4b, $p$-value $= 0.4$), both typically increased from ca. June until the end of the year. This was accompanied by a gradual decrease in WL. 2019, a year of severe drought, did however not fit this trend: both WL and $CH_4$ fluxes remained relatively low throughout the year, and the fluxes were decoupled from air temperature and WL. Microbial activity and $CH_4$ production scale with temperature[16], whilst soil water content, or WL, regulates several processes contributing to the net $CH_4$ flux measured above the surface: (1) it creates the anoxic conditions necessary for $CH_4$ production, (2) it decreases $CH_4$ oxidation by reducing the amount of oxygen available to methanotrophic microbial communities, and (3) it impacts the efficiency of the diffusion of $CH_4$ out of the soil. The combined effect of these terms, which operate in parallel, determines the magnitude and sign of the net surface-atmosphere flux of $CH_4$. The Okavango Delta receives pulsed seasonal flooding leading to alternating periods of soil wetting (April-August) and drying (September-March) in some parts the Delta; this hydrological regime impacts the balance of $CH_4$ production, oxidation and diffusion terms and thereby the magnitude of the net fluxes that alternate between periods of net emissions of $CH_4$ to the atmosphere, and periods of low emissions or even net oxidation, which we have observed at Nxaraga seasonal floodplain. Our findings are consistent with published experimental results and modelling approaches of wetland $CH_4$, which have established water table depth as a dominant control of emissions[32,44,45].

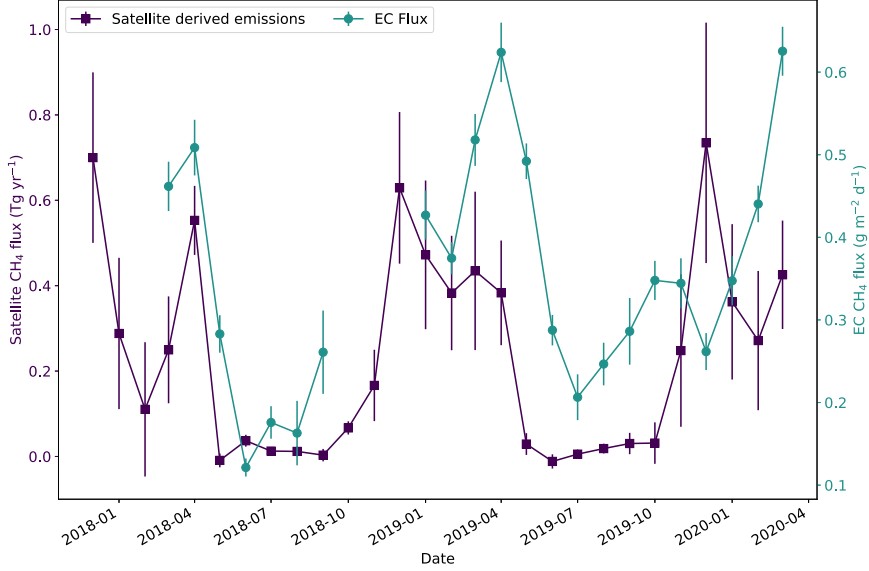

**Fig. 3 Local and delta-scale methane emission estimates.** Monthly emission estimates of $CH_4$ net fluxes ($g\,m^{-2}\,d^{-1}$) from eddy-covariance (EC) measurements (round symbols) over *Cyperus papyrus* at Guma Lagoon (18°57′53.01″S; 22°22′16.20″E), in the permanently flooded part of the Okavango Delta, and inferred from satellite observations of column $CH_4$ over the entire Okavango Delta ($Tg\,yr^{-1}$; square symbols). Data are presented as mean values ±2 standard deviations from the mean. The monthly EC budgets were constructed by summing diel cycles of hourly means; total uncertainty was obtained by propagating hourly standard deviation of the mean in quadrature ($n = 24$ independent hourly data points, see Eq. (1)). Due to the stochastic nature of the inversions used to derive emission estimates from satellite information, it is not possible to report a deterministic number of samples.

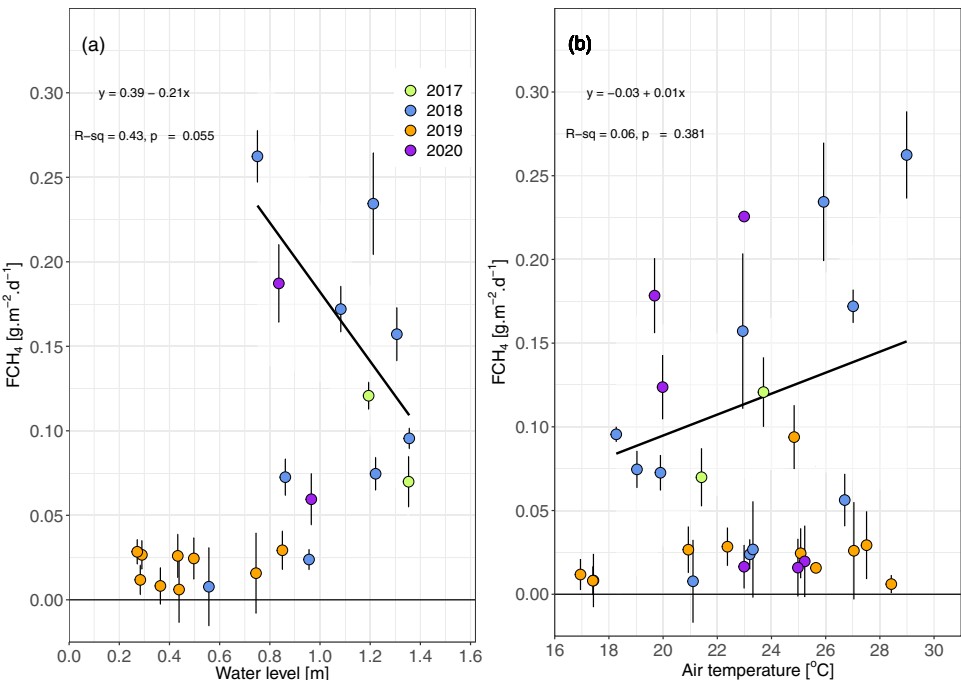

**Fig. 4 Relationship between monthly methane fluxes, water level and air temperature.** Monthly median $CH_4$ fluxes (FCH₄, $g\,m^{-2}\,day^{-1}$) ± inter-quartile range (IQR) from August 2017 to August 2020 as a function of (**a**) monthly water level, and (**b**) mean air temperature at the seasonal floodplain measurement site (19°32′53″S; 23°10′45″E). The number of half-hourly flux data points (*n*) from which median and IQR were calculated changed from month to month because of the variability of points filtered out by the micrometeorological quality control filter (see Methods). Consequently, *n* ranged from 33 (August 2020) to 570 (September 2019). The solid lines represent linear regressions on a temporal subset of the data (austral winter to summer 2017 and 2018) excluding the 2019 drought period (equations and t-test two-sided p-value given in the panels): standard error of (**a**) ±0.10 and 0.09, and (**b**) ±0.01 and 0.16 for slope and intercept, respectively.

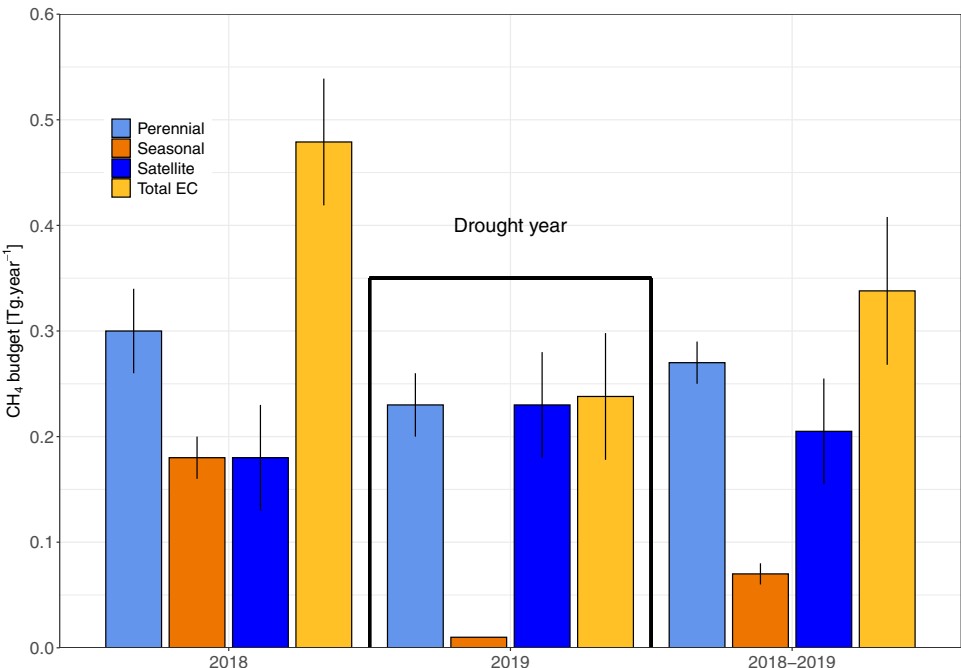

**Fig. 5 Annual CH₄ emission budgets by ecohydrological zones and for the entire Okavango Delta.** The budgets were obtained from upscaled eddy-covariance (EC) measurements and satellite observations. Total EC budgets are broken down into annual emissions from two hydrological zones (perennial and seasonal wetlands). The budgets for the occasionally flooded areas were negligible and were therefore left out. Individual budgets were constructed by summing the monthly emission estimates ($n = 12$) for each year. The error bars represent the uncertainty range for the respective emissions budgets; these were calculated by summing monthly uncertainties in quadrature ($n = 12$, see Methods and Eq. (1)).

**Methane flux upscaling from local to Delta scale**. We upscaled the measured emission budgets to the entire Okavango Delta by defining three ecohydrological areas with distinct CH₄ flux characteristics (Methods): (a) the papyrus swamp at Guma Lagoon was used as proxy for all permanently flooded areas, (b) the seasonal floodplain at Nxaraga was the proxy for all seasonally flooded areas, and (c) we assumed that net oxidation fluxes, as measured using closed chambers over dry sandy soil at the seasonal floodplain (see Gondwe et al.[46] for methodology), prevailed at the occasionally flooded (defined as flooded at least once per decade) areas throughout the year. Based on these assumptions, we estimate using our EC data that the Okavango Delta was a net annual source of CH₄ to the atmosphere of $0.48 \pm 0.09$ and $0.24 \pm 0.03$ Tg year$^{-1}$ in 2018 and 2019, respectively (Figs. 5, 6). The 2019 budget compares well with the independent emission estimate inferred from satellite observations ($0.23 \pm 0.05$ Tg year$^{-1}$). The 2018 EC budget was more than double the satellite estimate ($0.18 \pm 0.05$ Tg year$^{-1}$), which could be due to an overestimation of the contribution of the seasonal floodplain. The EC flux footprint entrains a highly heterogeneous landscape of seasonally flooded grasslands traversed by a major river, which means that the fluxes are not fully representative of the land area classified as seasonally flooded. Selecting only data points for which 90% of the flux originated from within 200 m from the EC tower to restrict the contributions from riparian and aquatic source areas, the local annual budget for Nxaraga is reduced by 34% in 2018 and 78% in 2019, but uncertainties are large ($0.12 \pm 0.05$ Tg in 2018 and $0.003 \pm 0.010$ Tg in 2019). In addition, we used maximum wetland extent to upscale local fluxes to the regional scale, but this merely approximates the seasonal dynamics because peripheral areas of the alluvial fan experience shorter flooding durations. As a result of these approximations, the upscaled budgets for the seasonally flooded wetlands are likely overestimated. Wetland extent was the dominant control of inter-annual variability in the upscaled budget for perennial swamps.

Whilst spatial uncertainties could not be quantified, the extent of the perennially flooded areas varies little at the seasonal timescale, and we therefore expect more robust upscaled CH₄ budgets. However, it must be also be noted that the uncertainties arising from using papyrus as a proxy for other major macrophyte communities (e.g. phragmites) are unknown. Were et al.[47], did not observe statistically significant differences in the magnitude and seasonality of soil CH₄ emissions between papyrus and phragmites plots in Uganda, but van den Berg et al. showed[48] that *Phragmites australis* possess strong diel cycles characterised by elevated emissions of CH₄ during daylight hours (a reverse emission cycle to that we observed in papyrus) during the growing season, and CH₄ venting through dead culms has been documented[49,50]. The percentage coverage of phragmites and papyrus in the Okavango Delta being unknown, the impact different emission mechanisms and patterns on upscaled fluxes cannot be estimated.

Using simplified assumptions, CH₄ emissions from the permanent, vegetated wetlands accounted for 63% and 97% of the overall CH₄ emission budget for the Okavango Delta during 2018 and 2019, respectively. These wetlands hence play a disproportionate role in the overall budget compared to their source area (24% of total extent).

We estimate that CH₄ emissions (4.8 Tg year$^{-1}$) from African *Cyperus papyrus* alone (0.1% land cover)[51] account for 6% of the continent's total CH₄ emissions (85 Tg year$^{-1}$)[12]. However, without considering seasonality in fluxes as observed at the perennial swamp, these estimates could range from 2% to 10% of the continent's budget (calculated using annual minimum and maximum emissions, respectively). This is a significant uncertainty range, equivalent to 6–30% of the inter-decadal global increase in CH₄ emissions from all sources (TD, 2010s compared to 2000s)[11]. This exemplifies the magnitude of the uncertainties on emissions from tropical vegetated wetland, and the urgency to better constrain them. This will require the development of a

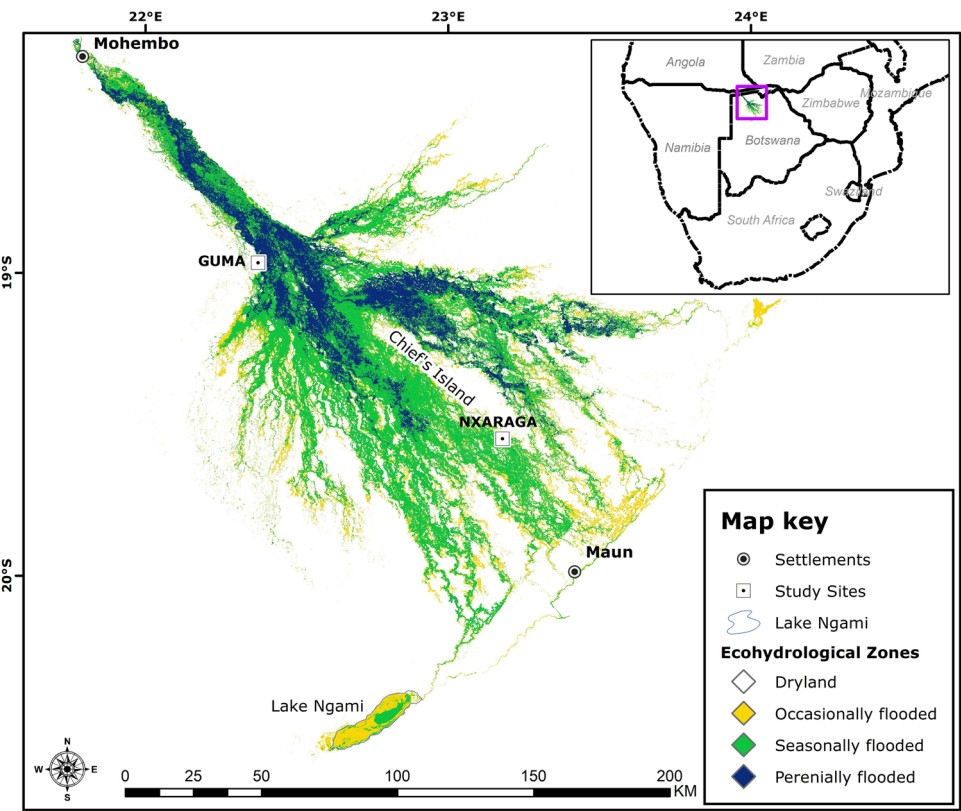

**Fig. 6 Ecohydrological zones of the Okavango Delta.** The map illustrates the spatial distribution and extent of the three main zones in 2019, based on a 25-year flood record and frequency-determined floodplain vegetation communities[71, 72].

detailed knowledge of the level of emission modulation performed by wetland vegetation globally, and identify direct observables (e.g. GPP) or proxies thereof (e.g. EVI) to upscale locally derived budgets and refine process-based models. In particular, understanding the vegetative controls on $CH_4$ production and oxidation, the environmental controls of plant-mediated transport at species level, and how climate and management impact them, will be key to forecasting future emissions of $CH_4$ in tropical wetlands.

## Methods

**Measurement sites**. The Okavango Delta in northern Botswana is one of the world's largest inland deltas with an estimated surface area of 40,000 $km^2$. Commonly termed delta, the Okavango is a low-gradient alluvial fan situated at the fringes of the Kalahari Desert[52,53]. Annual flooding occurs because of a pulsed discharge from the Cubango and Quito rivers, which originate in the Angolan Highlands and merge to form the Okavango River. The annual water influx from river discharge is estimated at 9 billion $m^3\,year^{-1}$ with a further 6 billion $m^3\,year^{-1}$ received as rainfall, predominantly during the austral summer months. Due to the low topographic gradient (1:3300), it takes the incoming floodwaters ca. 4–5 months to travel the 250 km separating the inlet at Mohembo from the main outlet at Maun, and 96–98% of the annual water input is lost through evapotranspiration, which is estimated at 1500 mm[54]. Peak flood extent occurs in August, and the extent of the annual inundation is controlled by the magnitude of the floodwater discharge and amount of rainfall, with evaporation playing a lesser role. The Okavango Delta can be divided into four physiographic zones: (a) an entry channel (the Panhandle), (b) a permanent swamp, (c) a seasonal swamp and, d) an occasional swamp.

The seasonal swamp areas are typically flooded 3–6 months per year, while flooding in the occasional swamp occurs at least once per decade. The delta is comprised of channels, wetlands and islands, which occur in varying proportions within the physiographic zones. Channels range from direct tributaries of the Okavango River to distributaries of the permanent swamps and outlets draining the perennial swamps[52]. Islands are dominated by non-aquatic vegetation ranging from trees to shrubs and grasslands; areas of bare, salt-crusted sandy soil are also found on islands, particularly in the interior. The flooding regime controls vegetation composition: reed grasses and sedges such as *Phragmites spp.* and *Cyperus papyrus* dominate the permanent swamps, whilst *Panicum repens* and *Oryza longistaminata* are typical in seasonal swamps[53,55,56].

We established two eddy-covariance (EC) measurement sites in the Okavango Delta: at Guma Lagoon (18°57′53.01″S; 22°22′16.20″E) in the permanent swamp area, and at Nxaraga (19°32′53″S; 23°10′45″E) in the seasonal swamp, on the SE edge of Chief's Island.

**Instrumentation**. The eddy-covariance instrumentation consisted of a Campbell Scientific IRGASON and a LI-COR 7700 open-path methane ($CH_4$) analyser. The IRGASON consists of a 3D ultrasonic anemometer and open-path infrared gas analyser, providing co-located measurements of the wind vector and mass densities of carbon dioxide ($CO_2$) and water vapour. The IRGASON was oriented into the prevailing wind direction at each site, and the LI-COR 7700 $CH_4$ analyser was mounted onto a horizontal boom, 0.3 m from the anemometer in the crosswind plane. A Vaisala WXT520 weather station recorded air temperature, pressure, relative humidity, wind speed and direction. Total solar radiation and photosynthetically active radiation (PAR) were measured by a Skye Instruments pyranometer (model SKS1110) and quantum detector (model SKP215), respectively. A Campbell Scientific CR3000 datalogger logged all sensors (sampling rate of 10 Hz for the EC variables and 10-s interval for the meteorological parameters) using a custom data acquisition program written in CR Basic (version 3.7).

At Guma lagoon, the instrumentation was mounted onto a 3-m high tripod, which was itself located on a 3-m high platform (effective measurement height 5.5 m). The EC system was installed on land, ca. 30 m to the west of a predominantly floating papyrus (*Cyperus papyrus*) mat. The papyrus mat was partially grounded to the West, where it met the shore of Guma Lagoon and it extended ca. 300 m into the lagoon in an easterly direction. The canopy height was on average 2.5 meters above water level.

At Nxaraga, a 2.5-m high EC mast was erected on the SW edge of Chief's Island to sample greenhouse gas (GHG) fluxes from the seasonal floodplain, which extends several hundred meters to the W, S and SW. The portion of floodplain within the flux footprint of the EC mast was bounded by a permanent, meandering water channel fringed by reeds and grasses such as *Phragmites spp* and *Miscanthus junceus*. The vegetation of the floodplain, which is dominated by grasses (e.g., *Panicum repens*, *Cynodon dactylon*, *Sporobolus spicatus*), attracts many types of herbivores (e.g. impala, buffalo) and is grazed for most of the year.

**Flux calculations and data quality control**. The EC data were processed into half-hourly fluxes using EddyPro® v.7.0.6. The eddy-covariance theory is well documented[57,58] and will not be discussed here. The core flux processing options

were raw data detrending using block averaging, double rotation of the wind vector (i.e. aligning the u-component streamwise and nullifying the vertical and crosswind components), application of the Webb–Penman–Leuning correction[59] to correct for density fluctuations, time lag determination using the covariance maximisation with default approach, and fully analytical methods of spectral corrections for low-[60] and high-pass[61] filtering effects.

Half-hourly flux data were rejected from further analysis if any of the following criteria was fulfilled:

- Failure of the micrometeorological data quality controls based on the assessment of steady state conditions and integral turbulence characteristics (flag value of 2, following the 0-1-2 flagging system proposed by Foken et al.[62,63]).
- Friction velocity ($u_*$) $<0.2$ m s$^{-1}$.
- Signal strength of the LI-COR 7700 open-path $CH_4$ analyser $<10\%$.
- Carbon dioxide fluxes outside the range $[-40, 40]$ μmol m$^{-2}$ s$^{-1}$.
- Methane fluxes outside the range $[-50, 1000]$ nmol m$^{-2}$ s$^{-1}$.
- Latent or sensible heat fluxes outside the range $[-250, 800]$ W m$^{-2}$.
- Wind blowing from outside the sector $[60°, 170°]$ at Guma Lagoon and $[100°, 170°]$ at Nxaraga.

**Monthly and annual budgets**. For each month of the study, half-hourly data were aggregated into hourly bins to construct 24-hour mean and median cycles of $CO_2$ and $CH_4$ fluxes. The uncertainty on each hourly mean and median data point was taken as the standard deviation and inter-quartile range, respectively. Aggregating data into hourly bins ensures that each of the 24 hourly points of the monthly diel cycle had the same weighting. This reduces the risk of biasing higher temporal flux statistics (e.g. daily, monthly or annual budgets) towards daytime values, as night time points are more likely to fail the data quality control criteria.

Daily budgets were calculated as the sum of hourly values and the associated uncertainties ($\sigma_{\text{day}}$) were obtained using standard error propagation rules (Eq. (1); $\sigma_i$ denotes the uncertainty on the flux value at hour $i$, with $i$ ranging from 0 to 23).

$$\sigma_{\text{day}} = \sqrt{\sum_{i=0}^{23} \left(\sigma_i\right)^2} \qquad (1)$$

Monthly budgets and uncertainties were calculated by multiplying the daily values by the number of days in a typical year (365) and dividing by 12; annual budgets were obtained by summing the monthly values. Following error propagation rules, monthly uncertainties were summed in quadrature as in Eq. (1) to obtain the total annual uncertainty.

**Carbon dioxide flux partitioning**. The fluxes of $CO_2$ were partitioned into ecosystem respiration ($R_{\text{eco}}$) and gross primary production (GPP) following the procedure of Saito et al.[64]. For $R_{\text{eco}}$, a two-step procedure was applied[65]:

- A non-linear function of temperature (Eq. (2)) was fitted to night time fluxes of $CO_2$.
- Assuming that the temperature dependency also extends to daytime, $R_{\text{eco}}$ was calculated for all available half-hourly time points using the non-linear parameterisation (Eq. (2); with $A$ and $B$ being fitting coefficients) on temperature ($T$) obtained with night time data.

$$R_{\text{eco}} = A exp^{BT} \qquad (2)$$

GPP was calculated as the difference between $R_{\text{eco}}$ and measured net $CO_2$ flux ($F_{CO2}$; Eq. (3)):

$$\text{GPP} = R_{\text{eco}} - F_{CO2} \qquad (3)$$

Maximum monthly gross primary production ($\text{GPP}_{\text{max}}$) was obtained by fitting the hyperbolic function of the photosynthetically active radiation (PAR) given in Eq. (4)[66] to the values of GPP calculated using Eq. (3).

$$\text{GPP} = \frac{\text{GPP}_{\text{max}}.\alpha.\text{PAR}}{\text{GPP}_{\text{max}} + \alpha.\text{PAR}} \qquad (4)$$

Temporal data aggregation, $CO_2$ flux partitioning and plotting were done using R version 4.0.3.

**Mapping of the ecohydrological zones of the Okavango Delta**. Zones with different flood regimes for the Delta have previously been mapped based on a combination of interpretation of satellite imagery, analogue aerial photography, ground-truthing and rule-based modelling[67–70]. These zone maps are however dated and of relatively low spatial resolution. Given the dynamic nature of the system, for this study we produced a new distribution map of ecohydrological zones based on a recent time-series of higher resolution remote sensing (Fig. 5) and statistically determined plant communities[71]. The communities were aggregated into larger groups for the purposes of this study, in which we needed to distinguish between perennially flooded areas and floodplains that experience regular seasonal flooding.

**Table 1 Annual extent of the three main ecohydrological zones in the Okavango Delta.**

| Year | Permanent [km$^2$] | Seasonal [km$^2$] | Occasional [km$^2$] | Total [km$^2$] |
|------|-----------|----------|------------|--------|
| 2018 | 2575 | 4923 | 2243 | 9741 |
| 2019 | 1911 | 1497 | 5669 | 9077 |

Mapping of the annual flood frequency in the Delta was based on a maximum inundation extent dataset derived from Landsat imagery[72] spanning the period 1990 to 2019. In five years of this sequence, one or more of the six annual images needed to produce the mosaicked composite image for the year were not available (1993, 2000, 2009, 2010 and 2012) and these years were excluded from the dataset. Frequencies are thus expressed as a fraction of the 25 time-step sequence, with each time step representing 4% of the total record. Frequency thresholds for transitions from one floodplain vegetation community to the next were identified through a combination of cluster analysis, indicator species analysis and species distribution modelling[73].

The areas of perennially, seasonally and occasionally inundated floodplain used for the upscaling are summarised in Table 1. The inflow in 2019 was the lowest on record (1934-present) and resulted in a contraction of the perennially and seasonally flooded area for that year, effectively increasing the area under occasional flooding. To capture this shift for upscaling, the annual perennially flooded area was subtracted from the total inundated area derived from high temporal resolution MODIS imagery[68,69].

**Upscaling of eddy-covariance fluxes**. An annual $CH_4$ budget for the entire Okavango Delta was calculated from the temporally and spatially weighted monthly budgets of the two measurement sites in the permanent and seasonal swamps (Eq. (5)). In addition, an annual budget for $CH_4$ oxidation in the occasional wetland was estimated from static chamber measurements of $CH_4$ fluxes over dry, sandy soil at Nxaraga. The dry soil chamber measurements were taken monthly (with a few exceptions) from August 2017 until August 2020 ca. 10–20 m inland from the EC mast overlooking the seasonal floodplain.

$$F_{\text{DELTA}} = \sum_{i=1}^{12} \left\{ A_P F_{P,i} + A_S F_{S,i} + A_O F_{O,i} \right\} \qquad (5)$$

In Eq. (5), $A_P$, $A_S$ and $A_O$ stand for the surface area of the permanent, seasonal and occasional wetlands, respectively, summarised in Table 1. $F_{P,i}$, $F_{S,i}$ and $F_{O,i}$ denote the mean methane flux during month number $i$ in the permanent, seasonal and occasional wetland, respectively.

**Determination of the enhanced vegetation index (EVI)**. We used SNAP (SNAP —ESA Sentinel Application Platform v8.0, http://step.esa.int) to estimate mean EVI values for the fraction of the papyrus stand within the flux footprint of the EC tower at Guma Lagoon (18°57′53.01″S; 22°22′16.20″E) from Sentinel-2 level 2A (bottom of atmosphere) reflectance imagery (tile number 34KFE). Where level 2A data was not available for download from the Copernicus Open Access Hub (http://scihub.copernicus.eu), we used the SNAP plugin Sen2Cor to process level 1C (top of atmosphere) data into level 2A. EVI was calculated for a set region of interest (Fig. 7) from Sentinel-2 spectral bands B2, B4 and B8 using Eq. (6).

$$\text{EVI} = 2.5*(B8 - B4)/(1 + B8 + 6*B4 - 7.5*B2) \qquad (6)$$

**Emission estimates inferred from satellite observations of $CH_4$**. We use $CH_4$ emission estimates from the Okavango Delta between December 2017 and March 2020 generated from satellite $CH_4$ column data from the TROPOspheric Monitoring Instrument (TROPOMI). The methods and estimates we follow those outlined in a recent study[74], which we briefly summarise here. The TROPOMI data were quality controlled for cloud coverage, surface albedo, aerosol optical depth and surface topography[73,74].

We use the GEOS-Chem chemistry transport model to relate prior surface emission estimates to atmospheric $CH_4$ concentrations and ultimately to TROPOMI $CH_4$ column data. The model was run in a nested configuration with a high-resolution simulation at $0.25° \times 0.3125°$ in the domain 36°S to 20°N and 20°W to 55°E. The concentrations at the edge of this regional domain were informed by the corresponding global model simulation with a coarser horizontal resolution of $2° \times 2.5°$ and empirically fitted to satellite data outside of the regional domain to ensure realistic model $CH_4$ concentrations at the lateral boundary conditions of the regional domain.

To infer the distribution and uncertainty of $CH_4$ emissions, we use an Ensemble Kalman Filter (EnKF) inverse method. A prior model simulation was driven each month by independent emission inventory estimates, including from wetlands, anthropogenic sources and biomass burning. To directly compare the GEOS-Chem model and TROPOMI data, we sample the high-resolution model

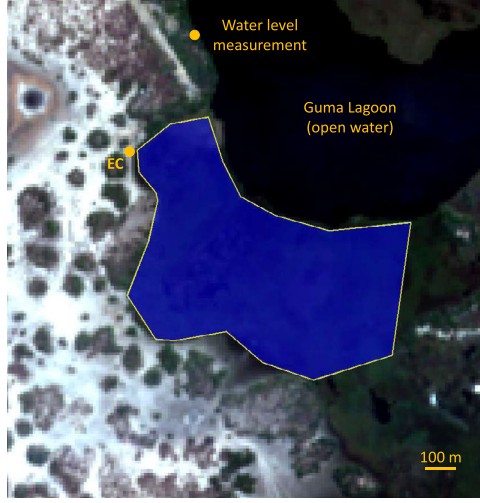

**Fig. 7 RGB Sentinel-2 imagery of the area surrounding the eddy-covariance (EC) tower at Guma Lagoon (18°57′53.01″S; 22°22′16.20″E).** The region of interest, from which pixels were sampled to calculate the enhanced vegetation index (EVI) of the floating papyrus stand, is shown as a blue polygon. The approximate location of the water level measurement sensor is also indicated.

CH$_4$ concentration distribution at the local time and location of individual TROPOMI scenes and convolve with scene-specific averaging kernels that describe the instrument vertical sensitivity to changes in CH$_4$. For our EnKF calculations we used 140 ensemble members, an assimilation window of 15 days, and a lag period of 1 month. Monthly CH$_4$ emission estimates from the Okavango delta were calculated as the sum of emissions inside a domain of 18.25°S–19.75°S and 22°E–24°E, a region covering the delta. Prior emissions in this region averaged 0.1 Tg yr$^{-1}$, of which over 90% were from wetlands during austral summer. Therefore, the posterior total is likely to be largely representative of wetland CH$_4$ emissions, as opposed to other sources.

**Reporting summary**. Further information on research design is available in the Nature Research Reporting Summary linked to this article.

## Data availability
The eddy-covariance and meteorological data generated in this study for the Guma Lagoon perennial wetland have been deposited with the UK Environmental Information Data Centre[75].

The eddy-covariance and meteorological data generated in this study for the for Nxaraga seasonal floodplain have been deposited with the UK Environmental Information Data Centre[76].

Both datasets are publicly available under the terms of the Open Government License (https://www.nationalarchives.gov.uk/doc/open-government-licence/version/3/).

Sentinel-2 imagery was obtained from Copernicus Open Access Hub (http://scihub.copernicus.eu).

## Code availability
The data acquisition (Campbell Scientific CR Basic) and data analysis (R) codes used in this study can be requested from the corresponding author. The approaches being standard, we did not deem the codes to merit deposition in online repositories.

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

## Acknowledgements

C.H. and U.S. thank the UK Natural Environment Research Council for financial support under grant numbers NE/N015746/1 and NE/N015746/2. M.F.L. and P.I.P. gratefully acknowledge funding from the Methane Observations and Yearly Assessments (MOYA) project (NE/N015916/1) and the National Centre for Earth Observation funded by the National Environment Research Council (NE/R016518/1). The authors are grateful to Mr. Edwin Mosimanyana and Dr. Kelebogile Mfundisi for help with logistics and fieldwork planning. C.H. and U.S. wish to thank Mr. Emmanuel Kambato, Mr. Thebe Kemosedile, Mr. Kemmonye W. Khaneguba, Mr. Kaelo Makati, Mr. Boatametse B. Mogojwa, Mr. Ineelo J. Mosie and Mr. Wakongwa Toro for technical support and for sharing their knowledge of the flora and fauna of the Okavango Delta. The authors are indebted to Mr. Guy Lobjoit for providing accommodation and access to facilities at Guma Lagoon Camp throughout the project.

## Author contributions

C.H. led the analysis of the ground flux dataset and the writing of the manuscript. M.G., U.S. and C.H. contributed to field data acquisition. M. M.-H. and A.M. led the mapping of the ecohydrological zones of the Okavango Delta. M.F.L. and P.I.P. delivered the satellite inverse modelling of the regional methane emission budgets. All authors contributed to the writing of the manuscript.

## Competing interests

The authors declare no competing interests.
