## [Peer Review File · Nature Communications]

REVIEWER COMMENTS

Reviewer #1 (Remarks to the Author):

General Comments

I enjoyed reading this paper and congratulate the authors on this impressive and successful research effort. My overall opinion is that the paper addresses an important gap in bottom-up methane flux data from one of the largest freshwater wetlands in Africa and it demonstrates the potential for refining methane budgets using plant phenology data. However, the work does not make a strong case for plant transport as a key process that regulates methane emissions. I believe the paper will have more impact if the authors change the focus slightly to emphasize the insights they gained from this study about the need for multiple proxies when upscaling methane flux data, rather than focusing on plant transport as an under-appreciated mechanism.

The introduction makes a compelling case for the urgent need to resolve differences in bottom-up and top-down estimates of regional methane budgets from ecosystems, with which I wholeheartedly agree. This is particularly difficult in Africa which has extensive wetlands but little bottom-up data and even less high-quality eddy flux data. The statement that this is the longest continuous methane flux record of wetland methane fluxes in Africa is striking and reason enough for its publication. I also agree with the authors' assessment that transport terms are poorly represented in wetland greenhouse models at all scales.

What I found most striking is the need for different upscaling strategies depending on the hydrologic condition of the site. GPP was a strong predictor for methane emissions in continuously flooded sites, for which a remotely sensed metric related to phenology such as EVI or synchronous reflectance may be a good proxy; water level was a strong predictor in seasonally flooded sites under normal climatic conditions; and presumably neither of were good predictors in seasonally flooded areas during drought. After establishing a good seasonal relationship between EC fluxes and satellite-derived emissions for papyrus, the authors used these insights to upscale the eddy flux data to the entire Delta and compared that top-down (satellite) estimates with mixed results. It seems to me that the novel aspect of the paper is in the EC-informed scaling strategy and insights into why they need to be addressed in methane budgets. This is similar to the concluding paragraph of the paper but rests more on the scaling strategy and less on the singular issue of plant transport.

Specific Comments

1. I suggest presenting the data from Guma Lagoon first because the continuously flooded sites are most important in the Okavango Delta methane budget, the relationships are more straightforward (helpful in brief-format papers), and this is where the more novel insights on phenology appear.

2. The text on page 4, lines 1-11, reports that GPPmax is a better predictor of methane flux than EVI. Lines 19-27 speculate that methane emissions are more strongly correlated to GPPmax because GPPmax is a better proxy for plant methane transport. Their argument is that mature plants are responsible for most plant methane transport which is captured better by GPPmax than EVI, and that juvenile plants are poor transporters of methane which is captured better by EVI than GPPmax. This interpretation is too simplistic.

The most parsimonious explanation for the relationship with GPPmax is that it is a proxy for carbon (or electron donor) supply that supports methanogenesis, a process that is strongly carbon limited. This fact is well established in the literature with the papers cited below as just a small sample. This relationship between plant productivity and methane emissions holds across time scales that range from seasons to hours as in the present study. Though carbon supply is the best supported explanation for the GPP-CH₄ relationship, I agree with the authors that plant transport may also be important, but only at short time scales.

I agree that plant transport helps explain the mid-day decline in methane flux shown in Extended Data Figure 1, but with one caveat. The decline coincides with peak CO₂ uptake (GPP) suggesting there is a process other than carbon supply involved, which is most likely to be plant transport. The authors invoke stomatal closure as the mechanism by which methane flux declines, but this assumes methane escapes plants primarily through stomates. This may be true for the dominant

species in Okavango Delta, but the authors should cite published studies that document stomatal control of methane emissions from these species. Please also acknowledge that methane also escapes from locations other than stomates such as the base of stems (including dead stems) and that the diurnal patterns observed may be influenced by the mechanisms driving gas flux through the stems of these species. For example, it is well established that methane flux in *Phragmites australis* is controlled by thermally driven convective flow, which would not respond to stomatal closure the way the authors suggest.

The idea that plant transport (rather than methane production) explains seasonal variation in methane emissions is unlikely because any methane that does not escape through plants will still escape through bubble ebullition and would therefore be observed by the eddy flux system. In principle methane that passes through plants could be subject to more methane oxidation than emission through ebullition, but the direction of that effect is opposite the seasonal pattern observed here.

Finally, I will suggest two alternative explanations for the relatively poor correlation with EVI. First, GPP is measuring an important driver of methanogenesis (carbon limitation) directly while EVI is a proxy for GPP and thus a step removed from the process. Second, given that EVI observes only the portion of the plant above the water it underestimates greenness to an extent that varies with water level which adds noise to the proxy but not to the direct measurement of GPP.

3. The text on page 4 from lines 11-18 explaining cycles of senescence and recruitment are not useful in my opinion because it is well known that vegetation indices based on reflectance integrate the biomass of both green and brown plants.

4. The main point of the section of the Nxaraga seasonal floodplain is the complex relationship between water level and temperature control on methane emissions, which is both interesting and important for the scaling exercise. However, I was a little confused by the way this was presented.

I suggest replacing "...suggests the existence of a "pivot point" for which maximum CH₄ emissions were realised" with a more simple statement such as "...suggests that water level and temperature are strong predictors of FCH₄ until water levels drop below a critical threshold, at which point these variables uncouple from FCH₄." This statement is straightforward and makes the main point instantly clear, while the term "pivot point" is vague.

5. I was also confused by the X axis of Figure 1 because the author's reason for normalizing water level by temperature is not intuitive. Given that temperatures in the Delta are relatively stable compared to water levels, I suspect that there are also a strong relationship with water level alone, in which case the meaning of the figure would be readily clear. If this relatively simple approach is not sufficient, then I suggest the authors explain why (mechanistically) it is necessary to add temperature to improve the model.

6. It seems to me that the upscaled estimates for the Okavango Delta are the most important outcome of this paper, especially given the well-articulated argument in the introduction about the mismatch in bottom-up and top-down estimates in methane budgets. The BU and TD estimates matched well in one year but not in another, but the only explanation given is a phrase about over-estimation of the seasonal floodplain flux. I suspect this statement relates to the text on page 2, lines 40-44 about limitations in the Boro River water level data (text that could be deleted in my opinion), but that is not clear. I suggest that the authors elaborate further on this part of the discussion.

7. The final paragraph on page 7 is also important but a bit too brief. It is not clear how the estimates with and without seasonality were made or what "seasonality" means in this case. Since the text is on *Cyperus papyrus* only I assume this is seasonality in plant activity (e.g. GPP); however, it could also be related to seasonality in the Seasonal Floodplain areas which was highly variable across years. This is another place where a more detailed discussion would be useful.

8. Figure 4. I suggest removing the "Occasional" category because it is too small to appear in the graph but adds distracting white-space gaps between the other categories within each year.

Papers showing GPP control of methane emissions and convective flux in *Phragmites*:

Undergraff K., Bridgham S. D., Pastor J., and Weishampel P. H. C. (2001) Ecosystem respiration

response to warming and water-table manipulations in peatland mesocosms. *Ecol. Appl.* 11, 311–326.

Van Der Nat, FF.W., Middelburg*, J.J., Van Meteren, D. et al. 1998. Diel methane emission patterns from *Scirpus lacustris* and *Phragmites australis*. *Biogeochemistry* 41, 1–22.

Vann C. D. and Megonigal J. P. (2003) Elevated CO₂ and water depth regulation of methane emissions: comparison of woody and non-woody wetland plant species. *Biogeochemistry* 63, 117–134.

Whiting G. J. and Chanton J. P. (1993) Primary production control of methane emission from wetlands. *Nature* 364, 794–795.

Purposefully signed by Pat Megonigal

Reviewer #2 (Remarks to the Author):

Helpful review for *Nature Comm.*

Helpfer and colleagues present results from the Okavango Delta where they have setup two EC towers. One was placed in a seasonal floodplain and the other a permanent wetland. The permanent wetland is particularly interesting due to the dominance of papyrus. They demonstrate that the phenological cycle of the papyrus has impact upon the CH₄ flux from the swamp on a seasonal scale. They also note some diurnal variations due to stomatal closure. Lastly they take these measurements and use them to upscale for the entire Delta.

I find the work to be interesting but not suitable for *Nat. Comm.* I think this would fit nicely with many more specialized journals but I don't see much interest for non-specialists as the work does not present overly new information or processes. So while I don't think it is right for this journal, I definitely encourage the authors to pursue publication of this work. Records of wetland CH₄ from African wetlands are very important for the wetland modelling community.

A few comments:

page 1 Line 38: top-down is more than inferred from atm. CH₄. The inversion models are constrained by them but the inversion models are a major part of this so perhaps include mention that this is also a modelled product and not just observations.

p2 l8 - wetland extent as model parameter? Perhaps rethink the use of the term parameter here.

p2 l16-20 - this whole section makes the reader think that the results presented here would allow then the use of a proper mechanistic treatment of plant mediated transport, but that isn't the case. This paper doesn't provide directly mechanistic information that could be used for this (but does provide some general means like productivity relates to CH₄ transport perhaps) so perhaps consider toning this down. It tends otherwise to get a reader's hopes up.

p2 l33 - Great! Long records are very useful to the community.

p2 l36 - reconsider the units. This is the first paper I can recall that uses ton/km²/month. Try sticking with the usual so that people don't have to do the math to figure out how this compares to their favourite wetland.

Fig 1 - why the line? Not mentioned in the caption. Sorta perhaps in the text but unclear. Also, a small point, the graphics look like they are from the 1990's. Consider giving them a more modern look.

Fig 3 - why the completely different units? What about trying to make them comparable?

p6 l18 - who did the overestimate? Unclear as written.

p7 l4 - Is there any indication that the papyrus CH₄ fluxes should be increasing due to expanding coverage? enhanced productivity? etc. As written it seems to imply that this could be part of the increase from the 2010s over the 2000s but I didn't see any proof given.

Reviewer #3 (Remarks to the Author):

The manuscript 'Phenology is the dominant control of methane emissions in a tropical non-forested wetland' by Helpfer et al represent a substantial contribution to our understanding of tropical

wetland methane emissions. In particular they use long term continuous EC data from two nearby hydrologically different sites and positively correlate GPP and EVI to flux of CH₄. This data covers a poorly represented geographical region (Africa) which may largely contribute to the natural tropical global methane emissions. Furthermore, long-term seasonal bottom-up studies are rare and the authors try to determine the major seasonal drivers of methane emissions which are always complex. The manuscript provides some valuable and novel diurnal insights into the role of plant-mediated methane emissions which I feel could be explored a little more. The manuscript is generally well written, logically presented however there are some things missing that I have outlined below.

General comments:

Diurnal wetland methane measurements are rare. New technologies such as continuous measurements from EC flux towers help provide new insights. I think the authors could include some more interpretations about the difference between day versus night time CH₄ efflux. Their monthly averaged data appear to show fairly clear diurnal trends of lower daytime emission during photosynthesis periods, morning pulses (as night-time methane accumulation may be released from plants as they open their stomata after dawn) and nighttime values – which are rarely measured. Because of this, previous studies conducted during daytime only may have over/ under estimate annual and global emissions. Therefore the authors have an opportunity to provide information about the % emission from day vs night for future studies elsewhere. This may change during growing seasons? Are we over or under or over estimating global wetland methane emissions if only working during the daylight?

The authors reply upon GPP and EVI correlations to conclude that plant-mediated emissions dominate control of the methane flux. But the authors do not measure soil, open water or ebullition pathways. It would have been useful to have some kind of a control measurement site to compare vegetated areas of the wetlands versus open water sites at the least. Could the authors use any satellite data with resolution that could compare open water flux to vegetated regions to support their findings about that plant pathway?

Specific Comments:

Page 2. line 1 – provide a ref for this 10% increase.

Page 2. Line 6-7 – you should cite some plant-mediated methane works with your wetland species here as there has been plenty on Phragmites and Papyrus over the years.

Page 2. Line 13-14- Whilst it is true that plant mediated is poorly studied, diurnal wetland plant-mediated emissions are also poorly studied– provide a few more refs here

Page 3 – Line 23 – The authors mention both Phragmites and Papyrus species here but most discussion then focuses only Papyrus plants. Do the authors know the approximate coverage of each species for these wetlands? Both species of plant are well studied in the literature however Phragmites can die back during winter months leaving stands of dead woody culms in the wetland. These have been shown to act like methane conduits transporting gas from deeper sediment to the atmosphere (see works of Armstrong and Armstrong, 1991; Chanton et al., 2002). Brown culms of Phragmites stands would not contribute to GPP and EVI values but could still contribute methane. This may therefore be a limitation in the authors approach.

Page 4 – Lines 2 and 8. What statistical tests and regressions were used here? Provide details.

Page 5. As mentioned above, night time wetland CH₄ flux measurements are novel. Similar to the comment at line 8, previous diurnal research in Phragmites showed that methane peaks occurred at the start of each day. This may be due to methane accumulation in the rhizosphere each night that is released as a pulse once the sun rises and PP begins. I also feel that in addition to partial stomatal closure the authors discuss, the role of oxygen transportation from the wetland adapted plants to their roots and rhizosphere (and the water column) may enhance soil and water oxidation rates during the daytime – (shown in many studies), which may also explain some of the observed trends of lower methane emissions during the day time during primary productivity periods.

This is a section of the manuscript where the authors could also briefly discuss the daytime vs night time contribution of methane these wetlands and how previous work focused on only daytime may over or under estimate seasonal and annual emissions.

Page 6 – Line 11-13. Please show where methane soil oxidation measurement were measured on the Fig. 5 map. Also, more information is required in the methods about how the closed chamber methane uptake measurements performed. How many per month? where? what equipment/

precision? Chamber specifications? In situ, ex situ analysis? If using a portable methane analyser, what kind and what r^2 thresholds for the linear regressions were used etc?

Page 10. Line 27 – some background information of the soil types would be useful here too if available.

Page 11 – line 8-13 – whilst the information of elephants and hippos etc are interesting to read about I am not sure they are needed here or fit into the 'instrumentation' section. If disturbances from these animals disrupted measurement period, data or created unnatural ebullition or methane release events, changed the physical vegetation you were measuring near etc that were accounted for in the data interpretation/ data gaps then provide this information.

Page 14- as mentioned above – more information about the soil oxidation measurement methods and equipment are required.

Page 15. Another figure 1? Should this be figure 6? Where is this figure mentioned in the manuscript?

Figures:

Figure 1. The overlapping dates are confusing and hard to read especially in the dry years and need to be more carefully placed. Is the trend line significant? If so include the r^2 , p value and information on the figure and or caption about these including the statistical test. These are requirements anyway if published in a Nature journal.

Figure 2. Similar to above comment, add r^2 , p values and stats test information. Did the authors try add 95% confidence intervals to the regression line?

Figure 4. This would be interesting to show the oxidation rates in this graph as negative emissions bars. Perhaps separate each year with vertical dashed lines so interpretation is easier. You could also label the 'drought' year so that this is more intuitive.

Figure 5. Needs scale and north arrow. Add water level measuring site and maybe add the four physiographic zones. Add soil oxidation measurements locations.

Table 1. Although easily calculable, a 'total' areal extent column and adding % of total for each ecohydrological zones might be useful for the reader interpretation.

Extended data Figure 1 – I suggest adding dark shaded areas showing night time to each graph. This might make the diel interpretations easier (indicating the morning spikes of methane just after sunrise) and show how this changes throughout the year as sunlight duration changes. What is the July FCH₄ value at 19:00 during a whole month that doesn't appear to fluctuate much?

Maybe a hippo?

End of review.

REVIEWERS' COMMENTS

Reviewer #1 (Remarks to the Author):

General Comments

I enjoyed reading this paper and congratulate the authors on this impressive and successful research effort. My overall opinion is that the paper addresses an important gap in bottom-up methane flux data from one of the largest freshwater wetlands in Africa and it demonstrates the potential for refining methane budgets using plant phenology data. However, the work does not make a strong case for plant transport as a key process that regulates methane emissions. I believe the paper will have more impact if the authors change the focus slightly to emphasize the insights they gained from this study about the need for multiple proxies when upscaling methane flux data, rather than focusing on plant transport as an under-appreciated mechanism.

The introduction makes a compelling case for the urgent need to resolve differences in bottom-up and top-down estimates of regional methane budgets from ecosystems, with which I whole-heartedly agree. This is particularly difficult in Africa, which has extensive wetlands but little bottom-up data and even less high-quality eddy flux data. The statement that this is the longest continuous methane flux record of wetland methane fluxes in Africa is striking and reason enough for its publication. I also agree with the authors' assessment that transport terms are poorly represented in wetland greenhouse models at all scales. (1).

What I found most striking is the need for different upscaling strategies depending on the hydrologic condition of the site. GPP was a strong predictor for methane emissions in continuously flooded sites, for which a remotely sensed metric related to phenology such as EVI or synchronous reflectance may be a good proxy; water level was a strong predictor in seasonally flooded sites under normal climatic conditions; and presumably neither of were good predictors in seasonally flooded areas during drought. After establishing a good seasonal relationship between EC fluxes and satellite-derived emissions for papyrus, the authors used these insights to upscale the eddy flux data to the entire Delta and compared that top-down (satellite) estimates with mixed results. It seems to me that the novel aspect of the paper is in the EC-informed scaling strategy and insights into why they need to be addressed in methane budgets. This is similar to the concluding paragraph of the paper but rests more on the scaling strategy and less on the singular issue of plant transport.

Specific Comments

1. I suggest presenting the data from Guma Lagoon first because the continuously flooded sites are most important in the Okavango Delta methane budget, the relationships are more straightforward (helpful in brief-format papers), and this is where the more novel insights on phenology appear.

Agreed. The text pertaining to the permanent papyrus swamp at Guma Lagoon has been moved to the start of the results section.

2. The text on page 4, lines 1-11, reports that GPPmax is a better predictor of methane flux than EVI. Lines 19-27 speculate that methane emissions are more strongly correlated to GPPmax because GPPmax is a better proxy for plant methane transport. Their argument is that mature plants are responsible for most plant methane transport which is captured better by GPPmax than EVI, and that

juvenile plants are poor transporters of methane which is captured better by EVI than GPPmax. This interpretation is too simplistic.

The most parsimonious explanation for the relationship with GPPmax is that it is a proxy for carbon (or electron donor) supply that supports methanogenesis, a process that is strongly carbon limited. This fact is well established in the literature with the papers cited below as just a small sample. This relationship between plant productivity and methane emissions holds across time scales that range from seasons to hours as in the present study. Though carbon supply is the best supported explanation for the GPP-CH4 relationship, I agree with the authors that plant transport may also be important, but only at short time scales.

We thank the reviewer for providing this alternative/complementary explanation for the relationship between CH₄ fluxes and GPP. We have taken this comment on board and rewritten the paragraph discussing this point (new text in green):

“ The largest CH₄ emissions (3.1 – 19.4 ton.km⁻².month⁻¹) in the Okavango Delta are from the permanent swamp areas, dominated by emergent *Cyperus papyrus* and *Phragmites australis*.

Monthly median fluxes of CH₄ (F_{CH4}) measured at the Guma Lagoon perennial papyrus swamp (18°57'53.01"S; 22°22'16.20"E) are linearly and positively correlated (R² = 0.89, p-value from t distribution < 0.05) with monthly maximum gross primary production (GPP_{MAX}, Methods) as shown in Fig.1a. GPP_{MAX} represents the maximum photosynthetic potential of the papyrus stand, a measure of the vigour/health of the stand at a particular time and the strongest predictor for F_{CH4}. GPP_{MAX} and F_{CH4} tend to increase with air and water temperature but these relationships are not statistically significant. This is consistent with other studies, which documented significant correlations between CH₄ fluxes and GPP at monthly to seasonal timescales. At these timescales, the net CH₄ fluxes to the atmosphere are most likely modulated by the CH₄ production term, which is stimulated by carbon (C) deposition in the rhizosphere from photosynthates^[31-33].

We also found a positive correlation between F_{CH4} and monthly estimates of enhanced vegetation index (EVI; Fig.1b, R² = 0.69, p-value < 0.05), and a comparable correlation (R² = 0.66, p-value < 0.05) between GPP_{MAX} and EVI. EVI, a proxy for leaf phenology or biomass (4), was obtained from Sentinel-2 imagery for the portion of the papyrus stand within the flux footprint of the EC tower. GPP_{MAX} is the strongest predictor for F_{CH4}, followed by EVI (R² = 0.54, p-value < 0.05). F_{CH4} and GPP_{MAX} tend to increase with air and water temperature but these relationships are not statistically significant. The phenological cycle of papyrus is complex: instead of a seasonal full stand die-back, senescence and recruitment of new shoots co-occur throughout the year, but the ratio of mature to senescing plants is variable [35]. The phenological cycle of papyrus inferred from EVI is characterised by high EVI values (green vegetation/ more aboveground biomass) during summer months (December-February) followed by a gradual decline (senescence) until winter (June-August). Greening broadly occurs from late winter onwards until next summer as water level decreases in the lagoon. We propose that GPP_{MAX}, a metric comparable to the photosynthetic capacity of Wu et al. (2016)^[34], is an indicator of the maturity of the papyrus stand or of leaf ontogeny. GPP_{MAX} might hence be more representative, or biased towards mature plants than EVI, because these are more productive/photosynthetically active than juvenile and senescent ones; as demonstrated by Wu et al. ^[34], leaf ontogeny is a strong predictor for GPP, but EVI cannot resolve leaf age, which results in a weaker correlation.

The correlation between F_{CH4} and EVI is likely an artefact of the circular dependency of F_{CH4} on GPP_{MAX} and GPP_{MAX} on EVI, but based on the available data it is not possible to rule out an element of seasonality in the plant-mediated transport term of CH₄. Indeed, a small study on papyrus reported

negligible CH₄ fluxes through the culms of juvenile and senescing individuals. By extrapolation, this could mean that the largest CH₄ fluxes for an entire papyrus stand occur when the proportion of mature plants reaches a maximum, i.e. when GPP_{MAX}, but not necessarily EVI, is at its maximum. This is consistent with the findings of Whiting and Chanton^[35], who explained the strong positive linear correlations between F_{CH₄} and biomass in a variety of plants, as plant-mediated transport. Working on the assumption that CH₄ is mainly lost through non-stomatal pathways such as pores in plant stems, they suggested that more biomass could equate with larger conduits for CH₄ transport and hence greater fluxes. In conclusion, whilst variations in physiology or plant-mediated transport efficiency are plausible, the dominant control of the seasonal cycle in CH₄ emissions is probably C-deposition to the rhizosphere.”

3. I agree that plant transport helps explain the mid-day decline in methane flux shown in Extended Data Figure 1, but with one caveat. The decline coincides with peak CO₂ uptake (GPP) suggesting there is a process other than carbon supply involved, which is most likely to be plant transport. The authors invoke stomatal closure as the mechanism by which methane flux declines, but this assumes methane escapes plants primarily through stomates. This may be true for the dominant species in Okavango Delta, but the authors should cite published studies that document stomatal control of methane emissions from these species. Please also acknowledge that methane also escapes from locations other than stomates such as the base of stems (including dead stems) and that the diurnal patterns observed may be influenced by the mechanisms driving gas flux through the stems of these species. For example, it is well established that methane flux in Phragmites australis is controlled by thermally driven convective flow, which would not respond to stomatal closure the way the authors suggest.*

These are very useful suggestions and we have extended the original text to address these points. New text is highlighted in green, and answers to specific comments are colour-coded to match the relevant reviewer’s suggestion above.

* The new text addressing this remark can be found under point 5.

“We observed marked diel cycles from October to March (Extended Data Fig. 1), characterised by a decrease in F_{CH₄} during the central daylight hours when vapour pressure deficit (VPD) was high and CO₂ uptake peaked. Overall, emissions of CH₄ were 52% ± 26 % larger at night than during the day (median ± inter-quartile range of monthly night to day ratios), and this night-to-day ratio increased with increasing EVI (Extended Data Fig.2). Both day and night time fluxes were linearly correlated with EVI but the night time slope was almost double the daytime one. Night time fluxes also had a weak statistically significant correlation with mean air temperature, but that was not the case for daytime fluxes. The implication of these differences between night and day fluxes is that estimating daily or higher temporal budgets from daytime measurements would lead to significant underestimations. To the best of our knowledge, there is no published literature on the dominant plant-mediated transport pathway for CH₄ in *Cyperus papyrus*, but a limited study documented constant CH₄ efflux from mature culms throughout daylight hours, and negligible emissions from juvenile and senescent plants (2). Such constant emission patterns do not fit the marked diel trends observed for most of the year, but stomatal flux regulation is a possibility. Jones and Muthuri^[37] showed that the stomatal conductance in papyrus canopies exhibits a sharp early morning rise followed by partial closure around midday as VPD increases. Partial stomatal closure reduces transpiration, even when root zone water availability is high, but has a lesser impact on photosynthesis in C4 plants^[38] such as papyrus^[39]. Consequently, partial stomatal closure could reduce CH₄ emissions as well as transpiration, while affecting CO₂ uptake, and by extension GPP_{max}, to a smaller degree. CH₄ fluxes through the umbels of the plants, which are comprised of hyperstomatal bracteoles and rays, and their diel cycles are however

unknown. Although plausible, stomatal control of CH₄ fluxes in papyrus at short timescales remain speculative.

Pressurisation of the internal lacunae found in many aquatic macrophytes in response to increasing VPD and, to a lesser degree, air temperature, can give rise to convective gas flow, albeit with temporal cycles opposite to the ones we measured over papyrus^[40,41]. Some authors have attempted to reconcile the concept of convective flow with observed trends of decreasing CH₄ emissions during daylight hours as venting of CH₄ accumulated overnight inside the plants until fluxes become limited by pressurised ventilation rather than stomatal conductance^[13,31,42]. To the best of our knowledge, pressurisation has not been studied in *Cyperus papyrus*, but other members of the Cyperaceae family (e.g. *C. involucratus* and *C. eragrostis*) are known not to produce significant convective flow^[40].

Diel cycles of O₂ fluxes through plants, and concentrations within the rhizosphere, which can decrease by 30% at night in papyrus as a result of metabolic O₂ demand and the absence of photosynthetic production^[43], offer an additional mechanism for the measured trends. A concomitant reduction in CH₄ oxidation at night could shift the balance of the CH₄ production and consumption terms towards a net increase in CH₄, giving rise to diel emission trends consistent with our measurements^[44]; This could also explain the strong correlation between CH₄ fluxes and EVI, particularly at night (Extended Data Fig.2), as more biomass (high EVI) could equate with higher metabolic O₂ demand and higher net CH₄ fluxes. In all likelihood, the trends in net CH₄ emissions result from a superposition of several processes, but our dataset cannot to resolve them.”

4. The idea that plant transport (rather than methane production) explains seasonal variation in methane emissions is unlikely because any methane that does not escape through plants will still escape through bubble ebullition and would therefore be observed by the eddy flux system. In principle methane that passes through plants could be subject to more methane oxidation than emission through ebullition, but the direction of that effect is opposite the seasonal pattern observed here.

We agree that the argument about the role of plant-mediated transport in the seasonality of the measured fluxes of CH₄ was too speculative. We have therefore toned down this argument and proposed C-limitation and CH₄ production as the overarching controls. The new text can be found under point 5 below.

5. Finally, I will suggest two alternative explanations for the relatively poor correlation with EVI. First, GPP is measuring an important driver of methanogenesis (carbon limitation) directly while EVI is a proxy for GPP and thus a step removed from the process. Second, given that EVI observes only the portion of the plant above the water it underestimates greenness to an extent that varies with water level which adds noise to the proxy but not to the direct measurement of GPP.

“Monthly median fluxes of CH₄ (F_{CH4}) measured at the Guma Lagoon papyrus swamp are linearly and positively correlated ($R^2 = 0.89$, p-value < 0.05) with monthly maximum gross primary production (GPP_{MAX}, Methods) as shown in Fig.1a. GPP_{MAX}, which represents the maximum photosynthetic potential of the papyrus stand, is a measure of the vigour/health of the stand at a particular time and the strongest predictor for F_{CH4}. F_{CH4} and GPP_{MAX} tend to increase with air and water temperature but these relationships are not statistically significant. This is consistent with other studies, which documented significant correlations between CH₄ fluxes and GPP at monthly to seasonal timescales. At such timescales, the net CH₄ fluxes to the atmosphere are most likely modulated by the CH₄ production term, which is stimulated by carbon (C) deposition in the rhizosphere from photosynthates^[31-33]. “

“The phenological cycle of papyrus is complex: instead of a seasonal full stand die-back, senescence and recruitment of new shoots co-occur throughout the year, but the ratio of mature to senescing plants is variable^[35]. The phenological cycle of papyrus inferred from EVI is characterised by high EVI values (green vegetation/ more aboveground biomass) during summer months (December-February) followed by a gradual decline (senescence) until winter (June-August). We propose that GPP_{MAX} , a metric comparable to the photosynthetic capacity of Wu et al. (2016)^[34] is an indicator of the maturity of the papyrus stand or of leaf ontology. GPP_{MAX} might hence be more representative (or biased towards) mature plants than EVI, because these are more productive/photosynthetically active than juvenile and senescent ones; as demonstrated by Wu et al.^[34], leaf ontology is a strong predictor for GPP, but EVI cannot resolve leaf age which results in a weaker correlation.

The correlation between F_{CH_4} and EVI is likely an artefact of the circular dependency of F_{CH_4} on GPP_{MAX} and GPP_{MAX} on EVI, but based on the available data it is not possible to rule out an element of seasonality in the plant-mediated transport term of CH_4 . Indeed, a small study on papyrus reported negligible CH_4 fluxes through the culms of juvenile and senescing individuals. By extrapolation, this could mean that the largest CH_4 fluxes for an entire papyrus stand occur when the proportion of mature plants reaches a maximum, i.e. when GPP_{MAX} , but not necessarily EVI, is at its maximum. This is consistent with the findings of Whiting and Chanton^[35], who explained the strong positive linear correlations between F_{CH_4} and biomass in a variety of plants, as plant-mediated transport. Working on the assumption that CH_4 is mainly lost through non-stomatal pathways such as pores in plant stems, they suggested that more biomass could equate with larger conduits for CH_4 transport and hence greater fluxes. In conclusion, whilst variations in physiology or plant-mediated transport efficiency are plausible, the dominant control of the seasonal cycle in CH_4 emissions is probably C-deposition to the rhizosphere.”

6. The text on page 4 from lines 11-18 explaining cycles of senescence and recruitment are not useful in my opinion because it is well known that vegetation indices based on reflectance integrate the biomass of both green and brown plants.

We removed “EVI is hence likely a measure of the ratio of green to brown plants.”, but kept the following text because the phenological cycle of papyrus might not be common knowledge:

“The phenological cycle of papyrus is complex: instead of a seasonal full stand die-back, senescence and recruitment of new shoots co-occur throughout the year, but the ratio of mature to senescing plants is variable [35]. The phenological cycle of papyrus inferred from EVI is characterised by high EVI values (green vegetation/ more aboveground biomass) during summer months (December-February) followed by a gradual decline (senescence) until winter (June-August).”

7. The main point of the section of the Nxaraga seasonal floodplain is the complex relationship between water level and temperature control on methane emissions, which is both interesting and important for the scaling exercise. However, I was a little confused by the way this was presented.

I suggest replacing “...suggests the existence of a “pivot point” for which maximum CH_4 emissions were realised” with a more simple statement such as “...suggests that water level and temperature are strong predictors of F_{CH_4} until water levels drop below a critical threshold, at which point the these variables uncouple from F_{CH_4} .” This statement is straightforward and makes the main point instantly clear, while the term “pivot point” is vague.

We have implemented the change suggested by the reviewer, and agree that the new wording improves the readability of the paragraph.

In addition, in the interest of brevity, we deleted the following sentences as per the reviewer's suggestion (point 6).

~~"Water level is was measured at the edge of the Boro River which bounds the floodplain, and it is hence only a proxy for average soil wetness/flooding in the flux footprint of the EC tower. In particular, there may exist an arbitrary, site-specific WL threshold below which soil wetness is no longer correlated to river level although CH₄ fluxes may remain sensitive to soil water content."~~

8. I was also confused by the X axis of Figure 1 because the author's reason for normalizing water level by temperature is not intuitive. Given that temperatures in the Delta are relatively stable compared to water levels, I suspect that there are also a strong relationship with water level alone, in which case the meaning of the figure would be readily clear. If this relatively simple approach is not sufficient, then I suggest the authors explain why (mechanistically) it is necessary to add temperature to improve the model.

We agree that the x-axis was not intuitive and have created 2 graphs to improve data visualisation. The revised Fig.1 is comprised of 2 panels: a) F_{CH₄} v water level, and b) F_{CH₄} v air temperature.

9. It seems to me that the upscaled estimates for the Okavango Delta are the most important outcome of this paper, especially given the well-articulated argument in the introduction about the mismatch in bottom-up and top-down estimates in methane budgets. The BU and TD estimates matched well in one year but not in another, but the only explanation given is a phrase about over-estimation of the seasonal floodplain flux. I suspect this statement relates to the text on page 2, lines 40-44 about limitations in the Boro River water level data (text that could be deleted in my opinion), but that is not clear. I suggest that the authors elaborate further on this part of the discussion.

The following paragraph has been added to expand the discussion:

"The EC flux footprint entrains a highly heterogeneous landscape of seasonally flooded grasslands traversed by a major river, which means that the fluxes are not fully representative of the land area classified as seasonally flooded. Selecting only data points for which 90% of the flux originated from within 200 m from the EC tower to restrict the contributions from riparian and aquatic source areas, the local annual budget for Nxaraga is reduced by 34% in 2018 and 78% in 2019, but uncertainties are large (0.12 ± 0.05 Tg in 2018 and 0.003 ± 0.010 Tg in 2019). In addition, we used maximum wetland extent to upscale local fluxes to the regional scale, but this merely approximates the seasonal dynamics because peripheral areas of the alluvial fan experience shorter flooding durations. Because of these approximations, the upscaled budgets for the seasonally flooded wetlands are likely overestimated. Wetland extent was the dominant control of inter-annual variability in the upscaled budget for perennial swamps. Whilst spatial uncertainties could not be quantified, the extent of the perennially flooded areas varies little at the seasonal timescale, and we therefore expect more robust upscaled CH₄ budgets."

10. The final paragraph on page 7 is also important but a bit too brief. It is not clear how the estimates with and without seasonality were made or what "seasonality" means in this case. Since the text is on *Cyperus papyrus* only I assume this is seasonality in plant activity (e.g. GPP); however, it could also be related to seasonality in the Seasonal Floodplain areas which was highly variable across years. This is another place where a more detailed discussion would be useful.

We have clarified these points as highlighted in the paragraph below:

“We estimate that CH₄ emissions (4.8 Tg.year⁻¹) from African *Cyperus papyrus* alone (0.1% land cover)^[40] account for 6% of the continent’s total CH₄ emissions (85 Tg.year⁻¹)^[12]. However, without considering seasonality in fluxes as observed at the perennial swamp, these estimates could range from 2% to 10% of the continent’s budget (calculated using annual minimum and maximum emissions, respectively), which is 6%-30% of the inter-decadal global increase in CH₄ emissions (TD, 2010s compared to 2000s)^[11]. This exemplifies the magnitude of the uncertainties on emissions from tropical vegetated wetland, and the urgency to better constrain them. This will require the development of a detailed knowledge of the level of emission modulation performed by wetland vegetation globally, and identify direct observables (e.g. GPP) or proxies thereof (e.g. EVI) to upscale locally-derived budgets and refine process-based models. In particular, understanding the vegetative controls on CH₄ production and oxidation, the environmental controls of plant-mediated transport at species level, and how climate and management impact them, will be key to forecasting future emissions of CH₄ in tropical wetlands. ”.

11. Figure 4. I suggest removing the “Occasional” category because it is too small to appear in the graph but adds distracting white-space gaps between the other categories within each year.

This has been implemented and the caption was amended to reflect this change:

“Figure 5: Annual CH₄ emission budgets for the Okavango obtained from upscaled eddy-covariance (EC) measurements and satellite observations. Total EC budgets are broken down into annual emissions from two hydrological zones (perennial and seasonal wetlands). The budgets for the

occasionally flooded areas were negligible and were therefore left out. The error bars represent the uncertainty range for the respective emissions budgets.”

Papers showing GPP control of methane emissions and convective flux in Phragmites:

Uppdergraff K., Bridgman S. D., Pastor J., and Weishampel P. H. C. (2001) Ecosystem respiration response to warming and water-table manipulations in peatland mesocosms. *Ecol. Appl.* 11, 311–326.

Van Der Nat, F.F.W., Middelburg*, J.J., Van Meteren, D. et al. 1998. Diel methane emission patterns from *Scirpus lacustris* and *Phragmites australis*. *Biogeochemistry* 41, 1–22.

Vann C. D. and Megonigal J. P. (2003) Elevated CO₂ and water depth regulation of methane emissions: comparison of woody and non-woody wetland plant species. *Biogeochemistry* 63, 117–134.

Whiting G. J. and Chanton J. P. (1993) Primary production control of methane emission from wetlands. *Nature* 364, 794–795.

Purposefully signed by Pat Megonigal (1)

Reviewer #2 (Remarks to the Author): Helfter review for Nature Comm.

Helfter and colleagues present results from the Okavango Delta where they have setup two EC towers. One was placed in a seasonal floodplain and the other a permanent wetland. The permanent wetland is particularly interesting due to the dominance of papyrus. They demonstrate that the phenological cycle of the papyrus has impact upon the CH₄ flux from the swamp on a seasonal scale. They also note some diurnal variations due to stomatal closure. Lastly they take these measurements and use them to upscale for the entire Delta.

I find the work to be interesting but not suitable for Nat. Comm. I think this would fit nicely with many more specialized journals but I don't see much interest for non-specialists as the work does not present overly new information or processes. So while I don't think it is right for this journal, I definitely encourage the authors to pursue publication of this work. Records of wetland CH₄ from African wetlands are very important for the wetland modelling community.

We thank the reviewer for their positive assessment, and we hope that the support for the manuscript voiced by the other reviewers and the editor will provide reassurance as to its suitability for publication in Nature Communications.

A few comments:

page 1 Line 38: top-down is more than inferred from atm. CH₄. The inversion models are constrained by them but the inversion models are a major part of this so perhaps include mention that this is also a modelled product and not just observations.

We have updated this line (new text in green) to reflect the more indirect nature of the observational constraint.

“Estimates of global emissions have large uncertainties, with bottom-up (BU) budgets (inferred from process-based models and inventories) overestimating top-down (TD) budgets (estimated through inversion modelling constrained by atmospheric CH₄ concentration measurements) by ca. 30%^[11].”

p2 l8 - wetland extent as model parameter? Perhaps rethink the use of the term parameter here.

After careful consideration we opted to keep the term parameter, because wetland extent is referred to as such in the scientific literature.

p2 l16-20 - this whole section makes the reader think that the results presented here would allow then the use of a proper mechanistic treatment of plant mediated transport, but that isn't the case. This paper doesn't provide directly mechanistic information that could be used for this (but does provide some general means like productivity relates to CH₄ transport perhaps) so perhaps consider toning this down. It tends otherwise to get a reader's hopes up.

The paragraph in question provides a snapshot of how plant-mediated transport, one of many model parameters, of CH₄ is handled in process-based models. Our intention is to provide a context for the state of understanding of CH₄ emissions from tropical aquatic wetlands.

Following the remarks from reviewer 1, the discussion of possible environmental drivers of CH₄ emissions at our study sites has been expanded and plant-mediated transport is but one mechanism,

and we acknowledge that it might not be the dominant one (see responses to reviewer 1 comments for further details of the changes made to interpretation and discussion of our results). We trust that these changes will remove the ambiguity the reviewer alluded to, but further changes can be made if requested.

p2 133 - Great! Long records are very useful to the community.

Many thanks for recognising the importance of these datasets.

p2 136 - reconsider the units. This is the first paper I can recall that uses ton/km2/month. Try sticking with the usual so that people don't have to do the math to figure out how this compares to their favourite wetland.

The units have been changed to $\text{g.m}^{-2}.\text{day}^{-1}$, which is widely used.

Fig 1 - why the line? Not mentioned in the caption. Sorta perhaps in the text but unclear. Also, a small point, the graphics look like they are from the 1990's. Consider giving them a more modern look.

At the request of reviewer 1, Fig.1 has been modified. The new version is now Fig.3 and is comprised of 2 panels showing FCH₄ as a function of water level and air temperature, respectively. The caption has been expanded to explain what the solid lines are (new text in green font below).

These plots were produced using the latest version of the ggplot2 R package. Style is intrinsically a matter of personal preference, and we found it difficult to determine what aspects were deemed old-fashioned. The other reviewers raised no such concern, and, in the absence of specific pointers, we opted to keep the original formatting.

Figure 3: Monthly median CH₄ fluxes ($\text{g.m}^{-2}.\text{day}^{-1}$) from August 2017 to August 2020 as a function of a) monthly water level, and b) mean air temperature at the seasonal floodplain measurement site (19°32'53" S; 23°10'45" E). **The solid lines represent linear regressions on a temporal subset of the data excluding the drought period (austral winter to summer 2017 and 2018).**

Fig 3 - why the completely different units? What about trying to make them comparable?

The units in Fig.2 (formerly Fig.3) are the native units for the quantities plotted: the satellite emissions are spatially integrated over the entire Okavango Delta, whereas the eddy-covariance data are fluxes expressed in units of mass per surface area and time ($\text{g}\cdot\text{m}^{-2}\cdot\text{d}^{-1}$). Our intention is to compare temporal trends between these 2 datasets, without introducing upscaling uncertainties into the eddy-covariance data. Satellite and eddy-covariance budgets for the entire Okavango Delta are shown side-by-side in Fig.4.

p6 l18 - who did the overestimate? Unclear as written.

This point was also raised by reviewer 1, and the discussion has been expanded to clarify it (new text in green):

“The 2018 EC budget was more than double the satellite estimate ($0.18 \pm 0.05 \text{ Tg}\cdot\text{year}^{-1}$), which could be due to an overestimation of the contribution of the seasonal floodplain. The EC flux footprint entrains a highly heterogeneous landscape of seasonally flooded grasslands traversed by a major river, which means that the fluxes are not fully representative of the land area classified as seasonally flooded. Selecting only data points for which 90% of the flux originated from within 200 m from the EC tower to restrict the contributions from riparian and aquatic source areas, the local annual budget for Nxaraga is reduced by 34% in 2018 and 78% in 2019, but uncertainties are large ($0.12 \pm 0.05 \text{ Tg}$ in 2018 and $0.003 \pm 0.010 \text{ Tg}$ in 2019). In addition, we used maximum wetland extent to upscale local fluxes to the regional scale, but this merely approximates the seasonal dynamics because peripheral areas of the alluvial fan experience shorter flooding durations. As a result of these approximations, the upscaled budgets for the seasonally flooded wetlands are likely overestimated. Wetland extent was the dominant control of inter-annual variability in the upscaled budget for perennial swamps. Whilst spatial uncertainties could not be quantified, the extent of the perennially flooded areas varies little at the seasonal timescale, and we therefore expect more robust upscaled CH_4 budgets.”

p7 l4 - Is there any indication that the papyrus CH_4 fluxes should be increasing due to expanding coverage? enhanced productivity? etc. As written it seems to imply that this could be part of the increase from the 2010s over the 2000s but I didn't see any proof given.

This sentence uses our study on papyrus to illustrate the importance of correctly quantifying the seasonality of CH_4 emissions from tropical wetlands, and puts these into a global context. We added the text in green below to emphasise that we are merely contextualising the uncertainty caused by improperly quantifying the seasonality of emissions.

“However, without considering seasonality in fluxes as observed at the perennial swamp, these estimates could range from 2% to 10% of the continent’s budget (calculated using annual minimum and maximum emissions, respectively). This is a significant uncertainty range, equivalent to 6%-30% of the inter-decadal global increase in CH_4 emissions from all sources (TD, 2010s compared to 2000s)^[11].”

Reviewer #3 (Remarks to the Author):

The manuscript 'Phenology is the dominant control of methane emissions in a tropical non-forested wetland' by Helfter et al represent a substantial contribution to our understanding of tropical wetland methane emissions. In particular they use long term continuous EC data from two nearby by hydrologically different sites and positively correlate GPP and EVI to flux of CH₄. This data covers a poorly represented geographical region (Africa) which may largely contribute to the natural tropical global methane emissions. Furthermore, long-term seasonal bottom-up studies are rare and the authors try to determine the major seasonal drivers of methane emissions which are always complex. The manuscript provides some valuable and novel diurnal insights into the role of plant-mediated methane emissions which I feel could be explored a little more. The manuscript is generally well written, logically presented however there are some things missing that I have outlined below.

We are grateful for this very positive assessment and for recognising the importance of this study. We have addressed all the comments listed below in italics, and new text (where applicable) is indicated in green.

General comments:

Diurnal wetland methane measurements are rare. New technologies such as continuous measurements from EC flux towers help provide new insights. I think the authors could include some more interpretations about the difference between day versus night time CH₄ efflux. Their monthly averaged data appear to show fairly clear diurnal trends of lower daytime emission during photosynthesis periods, morning pulses (as night-time methane accumulation may be release from plants as they open their stomata after dawn) and nighttime values – which are rarely measured. Because of this, previous studies conducted during daytime only may have over/ under estimate annual and global emissions. Therefore the authors have an opportunity to provide information about the % emission from day vs night for future studies elsewhere. This may change during growing seasons? Are we over or under or over estimating global wetland methane emissions if only working during the daylight?

The discussion of the diel cycles has been expanded (also at the request of reviewer 1) and a new figure was create (Extended Data Fig.2):

Supplementary Figure 1: Monthly median (all data years used) fluxes of CH₄ measured over a papyrus at Guma Lagoon (18°57'53.01"S; 22°22'16.20"E) during daytime (07:00 – 18:00, open symbols) and night time (19:00 – 06:00, solid symbols), as a function of monthly median enhanced vegetation index (EVI).

“We observed marked diel cycles from October to March (Extended Data Fig. 1), characterised by a decrease in F_{CH_4} during the central daylight hours when vapour pressure deficit (VPD) was high and CO₂ uptake peaked. Overall, emissions of CH₄ were 52% ± 26 % larger at night than during the day (median ± inter-quartile range of monthly night to day ratios), and this night-to-day ratio increased with increasing EVI (Extended Data Fig.2). Both day and night time fluxes were linearly correlated with EVI but the night time slope was almost double the daytime one. Night time fluxes also had a weak statistically significant correlation with mean air temperature, but that was not the case for daytime fluxes. The implication of these differences between night and day fluxes is that estimating daily or higher temporal budgets from daytime measurements would lead to significant underestimations. To the best of our knowledge, there is no published literature on the dominant plant-mediated transport pathway for CH₄ in *Cyperus papyrus*, but a limited study documented constant CH₄ efflux from mature culms throughout daylight hours, and negligible emissions from juvenile and senescent plants^[36]. Such constant emission patterns do not fit the marked diel trends observed for most of the year, but stomatal flux regulation is a possibility. Jones and Muthuri^[37] showed that the stomatal conductance in papyrus canopies exhibits a sharp early morning rise followed by partial closure around midday as VPD increases. Partial stomatal closure reduces transpiration, even when root zone water availability is high, but has a lesser impact on photosynthesis in C4 plants^[38] such as papyrus^[39]. Consequently, partial stomatal closure could reduce CH₄ emissions as well as transpiration, while affecting CO₂

uptake, and by extension GPP_{max} , to a smaller degree. CH_4 fluxes through the umbels of the plants, which are comprised of hyperstomatal bracteoles and rays, and their diel cycles are however unknown. Although plausible, stomatal control of CH_4 fluxes in papyrus at short timescales remain speculative.

Pressurisation of the internal lacunae found in many aquatic macrophytes in response to increasing VPD and, to a lesser degree, air temperature, can give rise to convective gas flow, albeit with temporal cycles in counter-phase with those measured over papyrus^[40,41]. Some authors have attempted to reconcile the concept of convective flow with observed trends of decreasing CH_4 emissions during daylight hours as venting of CH_4 accumulated overnight inside the plants until fluxes become limited by pressurised ventilation rather than stomatal conductance^[13,31,42]. To the best of our knowledge, pressurisation has not been studied in *Cyperus papyrus*, but other members of the Cyperaceae family (e.g. *C. involucratus* and *C. eragrostis*) are known not to produce significant convective flow^[40].

Diel cycles of O_2 fluxes through plants, and concentrations within the rhizosphere, which can decrease by 30% at night in papyrus as a result of metabolic O_2 demand and the absence of photosynthetic production^[43], offer an additional mechanism for the measured trends. A reduction in CH_4 oxidation at night could shift the balance of the CH_4 production and consumption terms towards a net increase in CH_4 , giving rise to diel emission trends consistent with our measurements^[44]; this could also explain the strong correlation between CH_4 fluxes and EVI, particularly at night (Extended Data Fig.2), as more biomass (high EVI) could equate with higher metabolic O_2 demand and higher net CH_4 fluxes. In all likelihood, the trends in net CH_4 emissions result from a superposition of several processes, but our dataset cannot to resolve them.”

The authors reply upon GPP and EVI correlations to conclude that plant-mediated emissions dominate control of the methane flux. But the authors do not measure soil, open water or ebullition pathways. It would have been useful to have some kind of a control measurement site to compare vegetated areas of the wetlands versus open water sites at the least. Could the authors use any satellite data with resolution that could compare open water flux to vegetated regions to support their findings about that plant pathway?

We acknowledge that plant-mediated transport is one possible mechanism, but likely not the dominant one. This point was also raised by reviewer 1 and we therefore proposed a number of mechanisms and processes which could explain the temporal cycles in CH_4 emissions we observed in the perennial papyrus swamp. The new text discussing processes compatible with the seasonal fluctuations are copied below, and possible diurnal controls were addressed in the previous comment.

We did not measure CH_4 fluxes over open water and we are not aware of any satellite data, which could be used to tease apart the contributions of open water and vegetation.

“Monthly median fluxes of CH_4 (F_{CH_4}) measured at the Guma Lagoon papyrus swamp are linearly and positively correlated ($R^2 = 0.89$, p -value < 0.05) with monthly maximum gross primary production (GPP_{MAX} , Methods) as shown in Fig.1a. GPP_{MAX} , which represents the maximum photosynthetic potential of the papyrus stand, is a measure of the vigour/health of the stand at a particular time and the strongest predictor for F_{CH_4} . F_{CH_4} and GPP_{MAX} tend to increase with air and water temperature but these relationships are not statistically significant. This is consistent with other studies, which documented significant correlations between CH_4 fluxes and GPP at monthly to seasonal timescales. At such timescales, the net CH_4 fluxes to the atmosphere are most likely modulated by the CH_4

production term, which is stimulated by carbon (C) deposition in the rhizosphere from photosynthates (7, 10, 11).

We also found a positive correlation between F_{CH_4} and monthly estimates of enhanced vegetation index (EVI; Fig.1b, $R^2 = 0.69$, p -value < 0.05), and a comparable correlation ($R^2 = 0.66$, p -value < 0.05) between GPP_{MAX} and EVI. EVI, a proxy for leaf phenology or biomass was obtained from Sentinel-2 imagery for the portion of the papyrus stand within the flux footprint of the EC tower. The phenological cycle of papyrus is complex: instead of a seasonal full stand die-back, senescence and recruitment of new shoots co-occur throughout the year, but the ratio of mature to senescing plants is variable^[35]. The phenological cycle of papyrus inferred from EVI is characterised by high EVI values (green vegetation/ more aboveground biomass) during summer months (December-February) followed by a gradual decline (senescence) until winter (June-August). We propose that GPP_{MAX} , a metric comparable to the photosynthetic capacity of Wu et al. (2016)^[34] is an indicator of the maturity of the papyrus stand or of leaf ontology. GPP_{MAX} might hence be more representative (or biased towards) mature plants than EVI, because these are more productive/photosynthetically active than juvenile and senescent ones; as demonstrated by Wu et al.^[34], leaf ontology is a strong predictor for GPP, but EVI cannot resolve leaf age which results in a weaker correlation.

The correlation between F_{CH_4} and EVI is likely an artefact of the circular dependency of F_{CH_4} on GPP_{MAX} and GPP_{MAX} on EVI, but based on the available data it is not possible to rule out an element of seasonality in the plant-mediated transport term of CH_4 . Indeed, a small study on papyrus reported negligible CH_4 fluxes through the culms of juvenile and senescing individuals. By extrapolation, this could mean that the largest CH_4 fluxes for an entire papyrus stand occur when the proportion of mature plants reaches a maximum, i.e. when GPP_{MAX} , but not necessarily EVI, is at its maximum. This is consistent with the findings of Whiting and Chanton^[35], who explained the strong positive linear correlations between F_{CH_4} and biomass in a variety of plants, as plant-mediated transport. Working on the assumption that CH_4 is mainly lost through non-stomatal pathways such as pores in plant stems, they suggested that more biomass could equate with larger conduits for CH_4 transport and hence greater fluxes. In conclusion, whilst variations in physiology or plant-mediated transport efficiency are plausible, the dominant control of the seasonal cycle in CH_4 emissions is probably C-deposition to the rhizosphere.”

Specific Comments:

Page 2. line 1 – provide a ref for this 10% increase.

The reference is given in the previous sentence.

Page 2. Line 6-7 – you should cite some plant-mediated methane works with your wetland species here as there has been plenty on Phragmites and Papyrus over the years.

We added the following references pertaining to CH_4 transport through Phragmites (Brix et al., 1992 & van den Berg et al., 2020) in a later part of the document, where we think they fit more naturally in the narrative. To the best of our knowledge, plant-mediated transport has not been studied in Papyrus, with the exception of one small study, which was referenced in the original version of the manuscript (Saunders, 2005).

Brix H, Sorrell BK, Orr PT. Internal Pressurization and Convective Gas-Flow in Some Emergent Fresh-Water Macrophytes. *Limnol Oceanogr.* 1992;37(7):1420-33.

van den Berg M, van den Elzen E, Ingwersen J, Kosten S, Lamers LPM, Streck T. Contribution of plant-induced pressurized flow to CH₄ emission from a Phragmites fen. *Scientific Reports.* 2020;10(1):12304.

Saunders, M.J., Fluxes of Carbon and Water in *Cyperus papyrus* L. tropical wetlands, in Department of Botany, 2005, University of Dublin, Trinity College: Dublin, Ireland. p. 233.

Page 2. Line 13-14- Whilst it is true that plant mediated is poorly studied, diurnal wetland plant-mediated emissions are also poorly studied– provide a few more refs here.

As part of the extended discussion of possible mechanisms consistent with the diel cycle we observed at the papyrus swamp, the following seminal papers were referenced:

Chanton JP, Whiting GJ, Happell JD, Gerard G. Contrasting Rates and Diurnal Patterns of Methane Emission from Emergent Aquatic Macrophytes. *Aquat Bot.* 1993; 46(2):111-28.

Whiting GJ, Chanton JP. Control of the diurnal pattern of methane emission from emergent aquatic macrophytes by gas transport mechanisms. *Aquat Bot.* 1996; 54(2-3):237-53.

Page 3 – Line 23 – The authors mention both Phragmites and Papyrus species here but most discussion then focuses only Papyrus plants. Do the authors know the approximate coverage of each species for these wetlands? Both species of plant are well studied in the literature however Phragmites can die back during winter months leaving stands of dead woody culms in the wetland. These have been shown to act like methane conduits transporting gas from deeper sediment to the atmosphere (see works of Armstrong and Armstrong,1991; Chanton et al., 2002). Brown culms of Phragmites stands would not contribute to GPP and EVI values but could still contribute methane. This may therefore be a limitation in the authors approach.

The following text (in green) was added to the discussion of the upscaled budgets to clarify the points raised by the reviewer:

“However, it must be also be noted that the uncertainties arising from using papyrus as a proxy for other major macrophyte communities (e.g. phragmites) are unknown. Were et al. (2021)^[48], did not observe statistically significant differences in the magnitude and seasonality of soil CH₄ emissions between papyrus and phragmites plots in Uganda, but van den Berg et al. (2016) showed that *Phragmites australis* possess strong diel cycles characterised by elevated emissions of CH₄ during daylight hours (a reverse emission cycle to that observed in papyrus) during the growing season^[49], and CH₄ venting through dead culms has been documented^[50,51]. The percentage coverage of phragmites and papyrus in the Okavango Delta being unknown, the impact different emission mechanisms and patterns on upscaled fluxes cannot be estimated.”

Page 4 – Lines 2 and 8. What statistical tests and regressions were used here? Provide details.

The sentence already states that the correlation is linear and we did hence not repeat that the regression was linear. The p-value was determined from a standard t distribution and this was added to the first instance of regression statistics listed in the text (see below).

“Monthly median fluxes of CH₄ (F_{CH4}) measured at the Guma Lagoon papyrus swamp are linearly and positively correlated ($R^2 = 0.89$, p-value from t distribution < 0.05) with monthly maximum gross primary production (GPP_{MAX}, Methods) as shown in Fig.1a.”

Page 5. As mentioned above, night time wetland CH₄ flux measurements are novel. Similar to the comment at line 8, previous diurnal research in Phragmites showed that methane peaks occurred at the start of each day. This may be due to methane accumulation in the rhizosphere each night that is released as a pulse once the sun rises and PP begins. I also feel that in addition to partial stomatal closure the authors discuss, the role of oxygen transportation from the wetland adapted plants to their roots and rhizosphere (and the water column) may enhance soil and water oxidation rates during the daytime – (shown in many studies), which may also explain some of the observed trends of lower methane emissions during the day time during primary productivity periods. This is a section of the manuscript where the authors could also briefly discuss the daytime vs night time contribution of methane these wetlands and how previous work focused on only daytime may over or under estimate seasonal and annual emissions.

We agree that multiple mechanisms are likely to be at play and we have expanded the discussion of possible controls (including the possible role of varying lacunar O₂ concentrations) at the request of reviewer 1. We therefore refer reviewer 3 to our responses to the comments of reviewer 1.

The comment regarding discussing differences between day and night fluxes was raised and addressed under the first point of reviewer 3' General comments.

Page 6 – Line 11-13. Please show where methane soil oxidation measurement were measured on the Fig. 5 map. Also, more information is required in the methods about how the closed chamber methane uptake measurements performed. How many per month? where? what equipment/precision? Chamber specifications? In situ, ex situ analysis? If using a portable methane analyser, what kind and what r2 thresholds for the linear regressions where used etc?

The soil oxidation measurements were conducted at Nxaraga (seasonal floodplain site marked on Fig.5). The scale of Fig. 5 is too large to resolve the location of that chamber measurement site, however exhaustive details including methodology and measurement protocols can be found in the following paper (accepted for publication in Philosophical Transactions A of the Royal Society and reference in the manuscript):

Gondwe et al., 2021, Methane flux measurements along a floodplain soil moisture gradient in the Okavango Delta, Botswana, doi: 10.1098/rsta.2020.0448.

Page 10. Line 27 – some background information of the soil types would be useful here too if available.

The 2 references listed in the previous sentence provide detailed information on soil types. In the interest of brevity, we elected not repeat this information.

Page 11 – line 8-13 – whilst the information of elephants and hippos etc are interesting to read about I am not sure they are needed here or fit into the 'instrumentation' section. If disturbances from these animals disrupted measurement period, data or created unnatural ebullition or methane

release events, changed the physical vegetation you were measuring near etc that were accounted for in the data interpretation/ data gaps then provide this information.

This sentence has been deleted. It was not possible to quantify the amount of disturbance caused by animal movements and hence no data filtering for such events was used.

Page 14- as mentioned above – more information about the soil oxidation measurement methods and equipment are required.

Comment acknowledged and answered above.

Page 15. Another figure 1? Should this be figure 6? Where is this figure mentioned in the manuscript?

This should indeed be Fig.6, thank you. Reference to this figure was added to methods section where it appears:

“EVI was calculated for a set region of interest (Fig.6) from Sentinel-2 spectral bands B2, B4 and B8 using Eq.6.”

Figures:

Figure 1. The overlapping dates are confusing and hard to read especially in the dry years and need to be more carefully placed. Is the trend line significant? If so include the r^2 , p value and information on the figure and or caption about these including the statistical test. These are requirements anyway if published in a Nature journal.

Fig.1 (now Fig.3 in revised manuscript) has been modified at the request of reviewer 1; it now contains two panels showing the relationship between methane fluxes and water level in Guma lagoon and air temperature, respectively. The overlapping data labels remain difficult to read for the drought year because of clustering; this clustering illustrates the insensitivity of methane fluxes to these environmental variable during the drought year, and we hope that this point comes across even if individual labels are hard to read. We experimented with colour- and shape-coding for month and year, but could not improve the readability of the plot.

Linear regressions and statistics were added to the plot panels as requested.

Figure 3: Monthly median CH₄ fluxes (g·m⁻²·d⁻¹) from August 2017 to December August 2020 as a function of a) monthly water level, and b) mean air temperature at the seasonal floodplain measurement site (19°32'53" S; 23°10'45" E). The solid lines represent linear regressions on a temporal subset of the data (austral winter to summer 2017 and 2018) excluding the 2019 drought period. The solid lines represent linear regressions (equations and regression statistics given in the panels).

Figure 2. Similar to above comment, add r^2 , p values and stats test information. Did the authors try add 95% confidence intervals to the regression line?

At the request of reviewer 1, Fig.2 has become Fig.1. As with the previous figure, we experimented with different display options, but concluded that replacing the text labels with e.g. colour-and shape-coding for month and year did not improve clarity.

Linear regressions and statistics were added to the plot panels as requested.

Figure 1: Mean monthly flux of CH₄ measured by EC over *Cyperus papyrus* at Guma Lagoon (18°57'53.01"S; 22°22'16.20"E) in the permanently flooded part of the Okavango Delta (August 2017 - August 2020) as function of a) maximum gross primary productivity (GPP_{MAX}), and b) enhanced vegetation index (EVI).

Figure 4. This would be interesting to show the oxidation rates in this graph as negative emissions bars. Perhaps separate each year with vertical dashed lines so interpretation is easier. You could also label the 'drought' year so that this is more intuitive.

The oxidation fluxes were too small to visualise on the scale of the graph. At the request of reviewer 1, fluxes from the occasional swamps (oxidation) were removed because they looked like empty spaces.

As requested, the drought year has been labelled explicitly on the bar chart (see new plot below).

Figure 4: Annual CH₄ emission budgets for the Okavango obtained from upscaled eddy-covariance (EC) measurements and satellite observations. Total EC budgets are broken down into annual emissions from two hydrological zones (perennial and seasonal wetlands). The budgets for the occasionally flooded areas were negligible and were therefore left out. The error bars represent the uncertainty range for the respective emissions budgets.

Figure 5. Needs scale and north arrow. Add water level measuring site and maybe add the four physiographic zones. Add soil oxidation measurements locations.

Scale and north arrow were added to the map as requested, and the eco-hydrological zones are colour-coded as described in the legend.

Water level was measured at the Guma site, and soil oxidation measurements were taken at Nxaraga. Both general locations are marked on the map, but the scale is too large to mark the exact sampling locations.

Instead, the water level measurement location at Guma Lagoon was added to Fig.6, and a new Extended Data Figure 4 was introduced to visualise the locations of the EC mast and dry soil chamber at Nxaraga seasonal floodplain.

Figure 5: Ecohydrological zones of the Okavango Delta in 2019, based on a 25-year flood record and frequency-determined floodplain vegetation communities^[72,73].

Figure 6: RGB Sentinel-2 imagery of the area surrounding the eddy-covariance (EC) tower at Guma Lagoon ($18^{\circ}57'53.01''\text{S}$; $22^{\circ}22'16.20''\text{E}$). The region of interest, from which pixels were sampled to calculate the enhanced vegetation index (EVI) of the floating papyrus stand, is shown as a blue polygon. The approximate location of the water level measurement sensor is also indicated.

Supplementary Figure 2: satellite image of the seasonal floodplain measurement site at Nxaraga (19°32'53" S; 23°10'45" E). The location of the eddy-covariance (EC) mast is indicated by an orange star, and a green star marks the location of the dry soil chamber used to measure methane oxidation fluxes. The orange polygon represents the median flux footprint of the EC mast.

Table 1. Although easily calculable, a 'total' areal extent column and adding % of total for each ecohydrological zones might be useful for the reader interpretation.

Change effected as requested.

Table 1: Annual extent of the three main ecohydrological zones in the Okavango Delta.

Year	Permanent [km ²]	Seasonal [km ²]	Occasional [km ²]	Total [km ²]
2018	2575	4923	2243	9741
2019	1911	1497	5669	9077

Extended data Figure 1 – I suggest adding dark shaded areas showing night time to each graph. This might make the diel interpretations easier (indicating the morning spikes of methane just after sunrise) and show how this changes throughout the year as sunlight duration changes. What is the July FCH4 value at 19:00 during a whole month that doesn't appear to fluctuate much? Maybe a hippo?

The plot has been amended as requested (see updated version below).

The cause for the July spike is unknown but animal activity is an option.

Extended Data Figure 1: Diel cycles of methane fluxes (FCH_4) measured by eddy-covariance at a permanently-flooded papyrus swamp at Guma Lagoon ($18^{\circ}57'53.01''\text{S}$; $22^{\circ}22'16.20''\text{E}$). Half-hourly methane flux data points were averaged to hourly values on a monthly basis. All data available for the period August 2017 – August 2020 were used. The coloured ribbon represents the standard deviation of the mean and the grey rectangles symbolise night time.

End of review.

REVIEWER COMMENTS

Reviewer #1 (Remarks to the Author):

I evaluated the rebuttal provided by Helfter and others on their manuscript "Phenology is the dominant control of methane emissions in a tropical non-forested wetland". I focused primarily on their response to my comments (reviewer 1) and found their revisions largely addressed my concerns. Specifically, the authors now advance carbon inputs as the most likely mechanism linking phenology to FCH₄, and they provide a more nuanced discussion of the role of plant transport that emphasizes the many uncertainties about the specific plant transport mechanisms that exist in the dominant species at their study sites.

My only remaining major concern is that the emphasis on plant carbon inputs as the primary mechanism by which phenology relates to FCH₄ is absent from the introduction, which instead is still focused on plant transport. Lines 16-25 of the revised manuscript discuss plant transport exclusively rather than the more general issue of phenology and its relationship multiple plant-mediated processes such as carbon inputs to soils (primary production), plant transport of CH₄ from soils, or plant transport of oxygen into soils. Revising this paragraph will take some thought because methane models do include plant production in some form.

Finally, I believe that lines 31-33 somewhat misrepresent the Whiting and Chanton paper by stating "This is consistent with the findings of Whiting and Chanton, who explained the strong positive linear correlations between FCH₄ and biomass in a variety of plants, as plant-mediated transport." It is true that Whiting and Chanton proposed plant-mediated transport as one explanation for their data, but they are quite clear that their primary hypothesis is carbon substrate inputs. This is evident in the title of the paper (Primary production control of methane emission from wetlands) and in the paragraph that precedes the one about plant transport:

"The relationship between CH₄ emission and NEP may be related to the stimulation of methanogenesis by the increase of substrates associated with recent production, including root exudation and turnover, and litter input. Variations in below-ground methane concentration correlated with plant presence in peatlands and the enhancement of CH₄ production and emission associated with rice plants indicate the importance of below-ground production. Results of laboratory experiments suggest the potential of exudation products to support methanogenesis."

So, I suggest they edit the sentence on L31-33 to: "This is consistent with the findings of Whiting and Chanton, who explained the strong positive linear correlations between FCH₄ and biomass in a variety of plants, as a combination of rates of the plant-mediated organic substrate supply and plant-mediated transport."

These remaining edits are relatively minor. My opinion is that the paper should be published in Nature Communications.

Reviewer #2 (Remarks to the Author):

Thank you authors for your revisions. I especially like the changes that resulted from reviewer #1's comments (who should be given acknowledgement in the MS acknow section I believe, given how important they were to the outcome).

Regarding my comments, I agree that plot aesthetics are purely a personal judgement, however please make it so that the labels of the data points don't overlap. Likely the easiest approach here is to use a colour scale for the dates and then remove the labels. Labels that are illegible are not useful - aesthetics aside. I see Rev #3 also wanted this addressed.

Given the directional change for the paper, I wonder if it may make sense to revise the intro section on page 2 starting at line 9. This is heavily talking about transport mechanisms and not phenology as the paper has pivoted. Perhaps a bit about phenology & transport would be useful if integrated?

I still generally feel that this paper might be more suited to a more specialized journal, but I see that I may be in the minority here and don't object to the enthusiasm of the other reviewers.

Reviewer #3 (Remarks to the Author):

The authors have done an excellent job in addressing my concerns and incorporating these in line with the other reviewers comments. Not further suggestions for me except a typo on page 3- line 2 "...but our dataset cannot to resolve them.".

REVIEWER COMMENTS

We thank the reviewers for their commitment and dedication to the peer-review process, and for their very positive assessments of the revisions made to the manuscript. We have addressed the remaining comments (in italics) as detailed in our point-by-point response below.

Reviewer #1 (Remarks to the Author):

I evaluated the rebuttal provided by Helfter and others on their manuscript "Phenology is the dominant control of methane emissions in a tropical non-forested wetland". I focused primarily on their response to my comments (reviewer 1) and found their revisions largely addressed my concerns. Specifically, the authors now advance carbon inputs as the most likely mechanism linking phenology to FCH₄, and they provide a more nuanced discussion of the role of plant transport that emphasizes the many uncertainties about the specific plant transport mechanisms that exist in the dominant species at their study sites.

My only remaining major concern is that the emphasis on plant carbon inputs as the primary mechanism by which phenology relates to FCH₄ is absent from the introduction, which instead is still focused on plant transport. Lines 16-25 of the revised manuscript discuss plant transport exclusively rather than the more general issue of phenology and its relationship multiple plant-mediated processes such as carbon inputs to soils (primary production), plant transport of CH₄ from soils, or plant transport of oxygen into soils. Revising this paragraph will take some thought because methane models do include plant production in some form.

We agree that too much emphasis was placed on discussing the role of plant-mediated transport of CH₄, and the modelling thereof. We altered the introduction to document other known processes and controls of net emissions of CH₄ to the atmosphere (new text in green), and shortened the discussion of the plant-mediated transport.

~~"Quantifying the role of non-woody wetland vegetation on regulating CH₄ transport to the atmosphere^{13, 14}.~~ This is particularly timely, given that revisions of conventional model parameters such as wetland extent¹³⁻¹⁵ and temperature-sensitivity of methanogenesis¹⁶ fail to reconcile BU and TD estimates or explain recent inter-annual variations in emissions. Net fluxes of CH₄ to the atmosphere result from complex and sometimes competing processes, which underpin production, oxidation and transport, and the magnitude and temporal dynamics of these terms are intimately linked to vegetative processes and growth cycles. For example, the availability of C for CH₄ production in soils, either from plant litter or photosynthates, can control CH₄ production and contribute to the modulation of daily to seasonal emissions in ecosystems spanning subarctic to subtropical latitudes^{17,18}, and the level of methane oxidation in soils has been shown to be larger in vegetated soils due to plant-mediated oxygenation of the rhizosphere^{19,20}. Plant-mediated transport of CH₄ can be the dominant transport mechanism²¹, but the efficiency of this pathway can be species-dependent and variable²²⁻²⁴.

~~Previous~~ Recent studies in the tropics have shown that trees can be substantial sources or pathways of CH₄ to the atmosphere^{25, 26}, but little less is known about the role of emergent macrophytes in non-forested tropical wetlands, which account for 20-37% of the global land surface of vegetated wetlands²⁷⁻²⁹. Whilst parameters such as soil C and inundation are commonly used in process-based models of CH₄ emissions, transport, and particularly plant-mediated transport of CH₄ is a relatively

poorly represented pathway. For example, five out of the ten wetland models evaluated in the WETCHIMP inter-comparison project^{24,25} did not include transport (e.g., ebullition, diffusion and plant mediation) as an emission pathway. In particular, the plant mediation transport term for emergent tropical macrophytes is largely an unknown quantity, and is typically described using temperate/boreal plant functional traits²⁵, by a generic conductance term²⁶ or treated implicitly by adjusting the amount of CH₄ production and oxidation in tropical wetland ecosystems²⁷⁻³⁰. **Regional and global estimates of wetland CH₄ emissions from models with and without explicit treatment of the transport pathway can vary by up to a factor two³⁰, and uncertainties on emissions from the data-poor tropics are particularly large.**"

Finally, I believe that lines 31-33 somewhat misrepresent the Whiting and Chanton paper by stating "This is consistent with the findings of Whiting and Chanton, who explained the strong positive linear correlations between FCH₄ and biomass in a variety of plants, as plant-mediated transport." It is true that Whiting and Chanton proposed plant-mediated transport as one explanation for their data, but they are quite clear that their primary hypothesis is carbon substrate inputs. This is evident in the title of the paper (Primary production control of methane emission from wetlands) and in the paragraph that precedes the one about plant transport:

"The relationship between CH₄ emission and NEP may be related to the stimulation of methanogenesis by the increase of substrates associated with recent production, including root exudation and turnover, and litter input. Variations in below-ground methane concentration correlated with plant presence in peatlands and the enhancement of CH₄ production and emission associated with rice plants indicate the importance of below-ground production. Results of laboratory experiments suggest the potential of exudation products to support methanogenesis."

So, I suggest they edit the sentence on L31-33 to: "This is consistent with the findings of Whiting and Chanton, who explained the strong positive linear correlations between FCH₄ and biomass in a variety of plants, as a combination of rates of the plant-mediated organic substrate supply and plant-mediated transport."

This is a very valid comment and we appreciate that taken in isolation this sentence ("This is consistent with the findings of Whiting and Chanton, who explained the strong positive linear correlations between FCH₄ and biomass in a variety of plants, as plant-mediated transport.") can be interpreted as misrepresenting the paper's main conclusion that primary production is the key control of CH₄ emissions. The original sentence referred to the paragraph of the cited paper following the one quoted by the reviewer:

"The relationship observed (Fig.1) may also be related to the action of plants providing conduits for gas exchange. In the process of aerating below-ground organs, plant ventilate methane. The apparent correlation between CH₄ emissions and live above-ground biomass (Fig.2) for most of these sites (notable exception: Typha, FL) is consistent with this conduit effect."

We thank the reviewer for suggesting an edit to the original sentence, which encapsulates both the primary and secondary hypotheses put forward by Whiting and Chanton. This revision has been incorporated in the new version of the manuscript.

“This is consistent with the findings of Whiting and Chanton¹⁸, who explained the strong positive linear correlations between F_{CH_4} and biomass in a variety of plants, as a combination of rates of the plant-mediated organic substrate supply and plant-mediated transport.”

These remaining edits are relatively minor. My opinion is that the paper should be published in Nature Communications.

Many thanks for this recommendation and for the transformative reviewing effort in both rounds of the process.

Reviewer #2 (Remarks to the Author):

Thank you authors for your revisions. I especially like the changes that resulted from reviewer #1's comments (who should be given acknowledgement in the MS acknow section I believe, given how important they were to the outcome).

Thank you for this comment and we agree that reviewer 1 deserves an acknowledgement. I do however believe that acknowledging reviewers is not permitted under Nature Communications editorial rules. The editor may be able to advise on this point?

The screenshot below was taken from Nat Comms formatting instructions document (<https://www.nature.com/documents/ncomms-formatting-instructions.pdf>)

ACKNOWLEDGEMENTS (optional)

Include funding sources.

Must be brief and must not include thanks to Editors or referees, effusive comments or dedications.

Regarding my comments, I agree that plot aesthetics are purely a personal judgement, however please make it so that the labels of the data points don't overlap. Likely the easiest approach here is to use a colour scale for the dates and then remove the labels. Labels that are illegible are not useful - aesthetics aside. I see Rev #3 also wanted this addressed.

The data labels in Figure 1 and 4 have been removed as requested, and years are represented by different colours. In the interest of readability, the data months were not colour- or shape-coded.

Figure 1: Mean monthly flux of CH₄ measured by EC over *Cyperus papyrus* at Guma Lagoon (18°57'53.01"S; 22°22'16.20"E) in the permanently flooded part of the Okavango Delta (August 2017 - August 2020) as function of a) maximum gross primary productivity (GPP_{MAX}), and b) enhanced vegetation index (EVI). The solid lines represent linear regressions (equations and regression statistics given in the panels).

Figure 4: Monthly median CH₄ fluxes (g.m⁻².day⁻¹) from August 2017 to August 2020 as a function of a) monthly water level, and b) mean air temperature at the seasonal floodplain measurement site (19°32'53" S; 23°10'45" E). The solid lines represent linear regressions on a temporal subset of the data (austral winter to summer 2017 and 2018) excluding the 2019 drought period.

Given the directional change for the paper, I wonder if it may make sense to revise the intro section on page 2 starting at line 9. This is heavily talking about transport mechanisms and not phenology as the paper has pivoted. Perhaps a bit about phenology & transport would be useful if integrated?

This point was also raised by reviewer 1, and the introduction was amended to rebalance the discussion of known controls of CH₄ emissions. Please, refer to the previous section for further details.

I still generally feel that this paper might be more suited to a more specialized journal, but I see that I may be in the minority here and don't object to the enthusiasm of the other reviewers.

Reviewer #3 (Remarks to the Author):

The authors have done an excellent job in addressing my concerns and incorporating these in line with the other reviewers comments. Not further suggestions for me except a typo on page 3- line 2 "...but our dataset cannot to resolve them."

We thank the reviewer for this positive assessment. The typo has been corrected.

REVIEWERS' COMMENTS

Reviewer #1 (Remarks to the Author):

I am fully satisfied with the revisions that I requested. Well done and best wishes.

REVIEWERS' COMMENTS

Reviewer #1 (Remarks to the Author):

I am fully satisfied with the revisions that I requested. Well done and best wishes.

We thank the reviewer for this recommendation.